# Oracle-Efficient Algorithms for
# Online Linear Optimization with Bandit Feedback[*]

**Shinji Ito**[†]
NEC Corporation, The University of Tokyo
i-shinji@nec.com

**Daisuke Hatano**
RIKEN AIP
daisuke.hatano@riken.jp

**Hanna Sumita**
Tokyo Metropolitan University
sumita@tmu.ac.jp

**Kei Takemura**
NEC Corporation
kei_takemura@nec.com

**Takuro Fukunaga**[‡]
Chuo University, RIKEN AIP, JST PRESTO
fukunaga.07s@g.chuo-u.ac.jp

**Naonori Kakimura**[§]
Keio University
kakimura@math.keio.ac.jp

**Ken-ichi Kawarabayashi**[§]
National Institute of Informatics
k-keniti@nii.ac.jp

## Abstract

We propose computationally efficient algorithms for *online linear optimization with bandit feedback*, in which a player chooses an *action vector* from a given (possibly infinite) set $\mathcal{A} \subseteq \mathbb{R}^d$, and then suffers a loss that can be expressed as a linear function in action vectors. Although existing algorithms achieve an optimal regret bound of $\tilde{O}(\sqrt{T})$ for $T$ rounds (ignoring factors of $\mathrm{poly}(d, \log T)$), computationally efficient ways of implementing them have not yet been specified, in particular when $|\mathcal{A}|$ is not bounded by a polynomial size in $d$. A standard way to pursue computational efficiency is to assume that we have an efficient algorithm referred to as *oracle* that solves (offline) linear optimization problems over $\mathcal{A}$. Under this assumption, the computational efficiency of a bandit algorithm can then be measured in terms of *oracle complexity*, i.e., the number of oracle calls. Our contribution is to propose algorithms that offer optimal regret bounds of $\tilde{O}(\sqrt{T})$ as well as low oracle complexity for both *non-stochastic settings* and *stochastic settings*. Our algorithm for non-stochastic settings has an oracle complexity of $\tilde{O}(T)$ and is the first algorithm that achieves both a regret bound of $\tilde{O}(\sqrt{T})$ and an oracle complexity of $\tilde{O}(\mathrm{poly}(T))$, given only linear optimization oracles. Our algorithm for stochastic settings calls the oracle only $O(\mathrm{poly}(d, \log T))$ times, which is smaller than the current best oracle complexity of $O(T)$ if $T$ is sufficiently large.

---
[*]This work was supported by JST, ERATO, Grant Number JPMJER1201, Japan.
[†]This work was supported by JST, ACT-I, Grant Number JPMJPR18U5, Japan.
[‡]This work was supported by JST, PRESTO, Grant Number JPMJPR1759, Japan.
[§]This work was supported by JSPS, KAKENHI, Grant Number JP18H05291, Japan.

# 1 Introduction

*Online linear optimization with bandit feedback*, or *bandit linear optimization*, is an important problem that has a wide range of applications. In it, a player is given $\mathcal{A} \subseteq \mathbb{R}^d$, referred to as a set of *action vectors*, and $T$, the number of *rounds* of decision-making. In each round $t \in [T] := \{1, 2, \ldots, T\}$, the player chooses an action $a_t \in \mathcal{A}$, and then observes loss $\ell_t^\top a_t$, where $\ell_t \in \mathbb{R}^d$ is an unknown *loss vector* that can change over rounds. The bandit linear optimization includes a variety of important online decision-making problems as special cases. For example, given a graph $G = (V, E)$ and $s, t \in V$, by setting $\mathcal{A} \subseteq \mathbb{R}^{|E|}$ to be the set of all characteristic vectors of $s$-$t$ paths, we can take into account *bandit shortest path* or *adaptive routing* [9]. In this setting, $\ell_t \in \mathbb{R}^{|E|}$ corresponds to (unknown) lengths of the edges, and bandit feedback $\ell_t^\top a_t$ represents the length of a chosen $s$-$t$ path $a_t$. In addition to this application, bandit linear optimization includes bandit versions of combinatorial optimization problems such as minimum spanning tree, minimum cut, and knapsack problem, as well as continuous optimization problems such as linear programming and semidefinite programming.

The performance of the player is evaluated in terms of *regret* $R_T(a^*)$, defined as $R_T(a^*) = \sum_{t=1}^T \ell_t^\top a_t - \sum_{t=1}^T \ell_t^\top a^*$ for $a^* \in \mathcal{A}$, which represents the difference between the cumulative loss for decision $\{a_t\}$ of the player and that for an arbitrarily fixed strategy $a^*$. The primary goal in bandit linear optimization is to achieve small regret for arbitrary $a^* \in \mathcal{A}$. Some existing algorithms achieve regret bounds of $\tilde{O}(\sqrt{T})$,[1] as shown in Tables 1 and 2. In contrast, papers [6; 8; 12; 21] showed that any algorithm will suffer at least $\Omega(\sqrt{T})$ regret in the worst case. Thus, algorithms with $\tilde{O}(\sqrt{T})$-regret bounds achieve *optimal* performance w.r.t. dependence on $T$.

Algorithms achieving an optimal $\tilde{O}(\sqrt{T})$-regret, however, have computational issues, especially when the action set $\mathcal{A}$ is exponentially large or is an infinite set. For example, well-known LinUCB methods [1; 16; 29] need to solve quadratic programming over $\mathcal{A}$, which has time complexity of $\Omega(|\mathcal{A}|)$ if there are no additional assumptions. The ComBand algorithm [11] runs efficiently if there is an efficient sampling algorithm for $\mathcal{A}$ (such as $k$-sets, spanning trees, or bipartite perfect matchings), but such sampling algorithms are open for many important examples, including $s$-$t$ paths. For the special case in which the convex hull of $\mathcal{A}$ can be expressed by $c$ linear inequalities, CombExp [13] runs in $O(\text{poly}(c, d)T)$-time. However, $c$ (the size of the linear inequality expression) can be exponentially large for many examples.

In this study, we aim to develop computationally efficient algorithms that achieve an $\tilde{O}(\sqrt{T})$ regret bound, under the assumption that we can call a *linear optimization oracle*. The oracle solves offline linear optimization problems over $\mathcal{A}$, i.e, given a loss vector $\ell \in \mathbb{R}^d$, the oracle outputs $a^* \in \arg\min_{a \in \mathcal{A}} \ell^\top a$. This assumption is standard in the context of online optimization [15; 23]. Under it, the computational efficiency of online optimization algorithms is evaluated in terms of *oracle complexity*: the number of oracle calls for the linear optimization oracle.

For online linear optimization with *full information*, in which a player can observe all entries of $\ell_t \in \mathbb{R}^d$ after choosing $a_t$, Kalai and Vempala [23] have proposed algorithms with an $\tilde{O}(\sqrt{T})$-regret bound and an oracle complexity of $O(T)$. Using this algorithm, McMahan and Blum [26] and Dani and Hayes [15] showed that one can achieve $\tilde{O}(T^{2/3})$-regret and $O(T^{1/2})$-oracle complexity for bandit linear optimization. However, it has been an open question as to whether or not we can achieve $\tilde{O}(\sqrt{T})$-regret and $\tilde{O}(\text{poly}(T))$-oracle complexity for bandit linear optimization, with only linear optimization oracles. In this study, we solve this open problem by proposing an algorithm that achieves $\tilde{O}(\sqrt{T})$-regret as well as $\tilde{O}(T)$-oracle complexity.

Here, we separately consider here two different settings for bandit linear optimization: a *non-stochastic setting* and a *stochastic setting*. In the non-stochastic setting, we do not assume any generative models, but $\ell_t$ may be chosen in an adversarial manner, depending on previous actions $a_1, \ldots, a_{t-1}$. The performance of an algorithm is measured in terms of the expectation of regret $R_T(a^*)$ w.r.t. the randomness of the algorithm's internal randomness and $\ell_t$. In the stochastic setting, by way of contrast, the loss vectors $\ell_t$ are assumed to follow a probability distribution $D$ over $\mathbb{R}^d$, i.i.d. for $t = 1, \ldots, T$.

## 1.1 Our Contribution

In this paper, we present computationally efficient algorithms that achieve $O(\text{poly}(d)\sqrt{T})$-regret. Specifically, we present algorithms with a small oracle complexity, i.e., algorithms that call the oracle as infrequently as possible. Our contribution is summarized in Tables 1 and 2.

For the non-stochastic setting, we propose an algorithm (Algorithm 1) that achieves $O(\sqrt{d^3 T \log T})$-regret in expectation and has $O(\text{poly}(d, \log T)T)$-oracle complexity.

**Theorem 1.** *For the non-stochastic setting, Algorithm 1 satisfies the following conditions:*

- *The output of the algorithm satisfies $\mathbf{E}[R_T(a^*)] = O(\sqrt{d^3 T \log T})$ for all $a^* \in \mathcal{A}$.*

- *The algorithm calls the linear optimization oracle $O(\text{poly}(d, \log T)T)$ times.*

- *The computational time, except for that of the oracle, is of $O(\text{poly}(d, T))$.*

As shown in Table 1, our Algorithm 1 achieves the smallest oracle complexity among algorithms with $\tilde{O}(\sqrt{T})$-regret. Noting that GeometricHedge assumes $\mathcal{A}$ to be a convex body, we can see that Algorithm 1 is the first algorithm that is applicable to discrete $\mathcal{A}$ and that achieves $\tilde{O}(\sqrt{T})$-regret and $\tilde{O}(\text{poly}(T))$-oracle complexity.

Although the first algorithm in Table 1 with $\tilde{O}(T^{2/3})$-regret and $O(T^{2/3})$-oracle complexity might look incomparable to our results, algorithms with such bounds can be easily constructed from our Algorithm 1. In fact, by dividing $T$ rounds into $T/B$ blocks of size $B > 1$ and regarding each block as an individual round, we obtain the following statement:

**Proposition 1.** *Suppose there exists an algorithm with $O(f(T))$-regret and $O(g(T))$-oracle complexity. Then, for arbitrary positive integer $B$, there exists an algorithm with $O(B \cdot f(T/B))$-regret and $O(g(T/B))$-oracle complexity.*

By setting the block size to be $B = \Theta(T^{1/3})$ and applying Algorithm 1, we can achieve $O(B\sqrt{T/B}) = \tilde{O}(T^{2/3})$-regret and $O(T/B) = \tilde{O}(T^{2/3})$-oracle complexity, which is equivalent to the the uppermost result in Table 1. Note that Proposition 1 does *not* lead to an $\tilde{O}(\sqrt{T})$-regret algorithm given an $\tilde{O}(T^{2/3})$-regret algorithm, conversely, since the block size $B$ must be at least 1.

For the stochastic setting, we propose an algorithm (Algorithm 2) that achieves $O(\sqrt{d^3 T \log(d \log T/\delta)})$-regret with probability $1 - \delta$ and has $O(\text{poly}(d, \log T))$-oracle complexity, where $\delta \in (0, 1)$ is an arbitrary parameter.

**Theorem 2.** *Suppose $\ell_t$ follows a distribution over $\mathbb{R}^d$, i.i.d. for $t = 1, 2, \ldots, T$. Algorithm 2 then satisfies the following conditions:*

- *The output of the algorithm satisfies $R_T(a^*) = O(\sqrt{d^3 T \log(d \log T/\delta)})$ for all $a^* \in \mathcal{A}$, with probability $1 - \delta$.*

- *The algorithm calls the linear optimization oracle $O(\text{poly}(d, \log T))$ times.*

- *The computational time, except for that of the oracle, is of $O(\text{poly}(d, T))$.*

A complete description of Algorithm 2 and a proof of this theorem are given in Appendix B. As shown in Table 2, all existing algorithms that achieve $\tilde{O}(\sqrt{T})$-regret require at least $\Omega(T)$ oracle complexity, and our Algorithm 2 is the first with an $\tilde{O}(\sqrt{T})$-regret bound and a sublinear oracle complexity in $T$.

In both Algorithms 1 and 2, we use the well-known techniques [30] of reduction among linear optimization, separation, and decomposition over a given convex body. Definitions of these three problems are given in Section 4. The reduction algorithms enable us to solve separation and decomposition problems by calling the linear optimization oracle $O(\text{poly}(d))$ times. Using these

Table 1: Regret Bound and Oracle Complexity of Non-Stochastic Bandit Linear Optimization

| Algorithm | Regret Bound | Oracle Complexity |
|---|---|---|
| MV algorithm [15; 26] with FPL [23] | $\tilde{O}(T^{2/3})$ | $O(T^{2/3})$ |
| ComBand [11], GeometricHedge [17], Exp2 [6] | $\tilde{O}(T^{1/2})$ | – |
| GeometricHedge with Volumetric Spanners[2][19] | $\tilde{O}(T^{1/2})$ | $\tilde{O}(T^7)$ |
| **Algorithm 1 [This paper]** | $\tilde{O}(T^{1/2})$ | $\tilde{O}(T)$ |

Table 2: Regret Bound and Oracle Complexity for Stochastic Bandit Linear Optimization

| Algorithm | Regret Bound | Oracle Complexity |
|---|---|---|
| LinRel [7], LinUCB with $\ell_2$-ball [1; 16; 29] | $\tilde{O}(T^{1/2})$ | – |
| LinUCB with $\ell_1$-ball [16], | $\tilde{O}(T^{1/2})$ | $O(T)$ |
| Linear Thompson sampling [2; 4] | $\tilde{O}(T^{1/2})$ | $O(T)$ |
| **Algorithm 2 [This paper]** | $\tilde{O}(T^{1/2})$ | $O(\mathrm{poly}(d, \log T))$ |

reduction techniques, Algorithms 1 and 2 maintain, respectively, supersets and subsets of the convex hull of $\mathcal{A}$ (=: $\mathrm{Conv}(\mathcal{A})$).

To construct Algorithm 1 for the non-stochastic setting, we extend a *cutting-plane approach* to our bandit-feedback setting. The cutting-plane approach, a way of reducing oracle complexity, has been applied only to full-information settings [20], not a bandit-feedback setting. A major difference between bandit-feedback and full-information settings is that the former needs *exploration*, i.e., chosen actions should be randomized with sufficiently large variance, whereas the latter does not need it and chooses actions deterministically. In full-information settings, hence, it suffices to focus on a deterministically chosen action alone. In contrast to this, to deal with the bandit-feedback setting, the difficulty lies in constructing a distribution of actions with sufficiently large variance, for which cutting planes can be efficiently computed and the number of them can be bounded.

To this end, we design relevant probability distributions so that the cutting-plane approach works, which successfully reduces oracle complexity. Specifically, the cutting-plane approach maintains convex bodies $\mathcal{K}_t$ that include and approximate $\mathrm{Conv}(\mathcal{A})$, from which we choose *candidates* for actions, employing the support of the probability distribution of actions to choose. It is only when some candidates are invalid, i.e., when some are outside of $\mathrm{Conv}(\mathcal{A})$, that $\mathcal{K}_t$ is updated with a cutting plane excluding such an invalid candidate. To bound the number of oracle calls, we design candidates for actions that satisfy two conditions: the set of candidates has a bounded cardinality, and each candidate is sufficiently close to the weighted center of $\mathcal{K}_t$. Thanks to the first condition, we can efficiently decide if invalid candidates exist. The second condition is essential for bounding the number of oracle calls in each update of $\mathcal{K}_t$.

Our Algorithm 2 for the stochastic setting is based on the framework of *phased elimination of actions*, in which $T$ rounds are divided into *phases* that are segments of consequent rounds, and, in each phase, actions are eliminated so that only promising ones are left. Existing works employing this framework [7; 24; 31] are computationally inefficient, mainly for the following two reasons: (i) We need to maintain a set of promising actions that may be an exponentially large combinatorial set, and, (ii) when choosing actions, we need to solve hard optimization problems, e.g., G-optimal design [24] or quadratic programming [7].

Our idea for resolving the first computational issue is to maintain the set of promising actions as a convex set instead of a subset of actions. The convex set here can be represented with only $O(\mathrm{poly}(d) \log T)$ linear inequalities, which implies that operations over it can be conducted efficiently. We resolve the second computational issue by combining *barycentric spanners* [9] and the decomposition technique over convex bodies [30], both of which are efficiently computed with $O(\mathrm{poly}(d))$ oracle calls. We show that, thanks to these techniques, we can estimate the loss vector with enough accuracy to achieve an $\tilde{O}(\sqrt{T})$-regret bound. The oracle complexity is bounded as follows: In our algorithm, all $a_t$ chosen in each phase are determined at the beginning of the phase, which means that the oracle complexity depends not on the number of rounds, but on the number of phases. The number of phases is of $O(\mathrm{poly}(d) \log T)$ and that of oracle calls in each phase is of $O(\mathrm{poly}(d, \log T))$, which results in overall $O(\mathrm{poly}(d, \log T))$-oracle complexity.

## 2 Related Work

For the *full-information setting* in which a player can observe $\ell_t \in \mathbb{R}^d$ rather than $\ell_t^\top a_t$, Follow the Perturbed Leader (FPL) by Kalai and Vempara [23] achieves $O(\sqrt{T})$-regret and $O(T)$-oracle complexity. This algorithm is used as a subroutine in MV algorithm [15; 26] (see Table 2).

For a more general problem referred to as *online improper learning*, in which only an approximate linear optimization oracle is given, Kakade et al. [22] have proposed the first efficient algorithms that achieve approximate regret of $O(\sqrt{T})$ for the full-feedback setting, and $O(T^{2/3})$ in the bandit-feedback setting. Recent papers by Garber [18] and Hazan et al. [20] have improved oracle complexity. Algorithms in [20] achieve oracle complexity of $\tilde{O}(T)$ in the full-feedback setting and $\tilde{O}(T^{2/3})$ in the bandit feedback setting with the same regret bound as in Kakade et al. [22]. For online improper learning with bandit feedback, however, constructing an efficient algorithm achieving $\tilde{O}(\sqrt{T})$ poses difficulties that have yet to be overcome.

In addition to the studies listed in Tables 1 and 2, there exist efficient algorithms for bandit linear optimization that work under different assumptions. Abernethy et al. [3] proposed a computationally efficient algorithm achieving $O(\sqrt{T})$-regret under the assumption that $\mathcal{A}$ is a convex body and that a self-concordant barrier [27] for $\mathcal{A}$ is given. However, constructing self-concordant barriers is not always possible with a linear optimization oracle alone, and, hence, this algorithm does not always work under our assumptions of linear optimization oracle and Assumption 1 given in the next section.

## 3 Problem Setting

The *bandit linear optimization problem* is a repeated game described as follows: Before the game starts, a player is given the number $T$ of rounds and the dimensionality $d$ of the *action set* $\mathcal{A} \subseteq \mathbb{R}^d$. In each round $t = 1, 2, \ldots, T$, the player chooses $a_t \in \mathcal{A}$ while an environment chooses a loss vector $\ell_t \in \mathbb{R}^d$, and then the player observes a loss $\ell_t^\top a_t$. The goal of the player is to achieve a small *regret* $R_T(a)$, which is defined for an arbitrary $a \in \mathcal{A}$ as $R_T(a) := \sum_{t=1}^T \ell_t^\top a_t - \sum_{t=1}^T \ell_t^\top a$.

We assume the action set $\mathcal{A}$ to be a compact set. Suppose that we have an algorithm for linear optimization over $\mathcal{A}$ for any vector $w \in \mathbb{R}^d$, which we call *linear optimization oracle* $\mathcal{O}_\mathcal{A} : \mathbb{R}^d \to \mathcal{A}$ that receives an input $w \in \mathbb{R}^d$ and returns a point $\mathcal{O}_\mathcal{A}(w) \in \mathcal{K}$ satisfying $w^\top \mathcal{O}_\mathcal{A}(w) = \min_{a \in \mathcal{A}} w^\top a$.

*Assumption* 1. We assume that there exist positive real numbers $L$ and $R$ such that (a) $\|\ell_t\|_2 \leq L$ for all $t \in [T]$, and (b) $\|a\|_2 \leq R$ for all $a \in \mathcal{A}$. In addition, we assume that (c) $\mathcal{K} := \mathrm{Conv}(\mathcal{A})$ has a positive volume, i.e., $\mathrm{Vol}(\mathcal{K}) := \int_\mathcal{K} 1 \mathrm{d}x > 0$.

The first two assumptions (a) and (b) are standard in bandit linear optimization. If we are given a linear optimization oracle over $\mathcal{A}$, we can assume (c) without loss of generality. In fact, if $\mathcal{A}$ is included in a subspace with a smaller dimension than $d$, we can then detect it by calling the linear optimization oracle polynomial times (see, e.g., Corollary 14.1g in [30]), and we can make $\mathcal{K}$ full-dimensional by ignoring redundant dimensions.

## 4 Preliminaries

### 4.1 Linear Optimization, Separation, and Decomposition

We define a *linear optimization problem* (LP), *separation problem* (SP), and *decomposition problem* (DP) for a compact convex body $\mathcal{P} \subseteq \mathbb{R}^d$ as follows:

*Problem* 1 (linear optimization problem, LP). Given a vector $w \in \mathbb{R}^d$, find a vector $x^* \in \mathcal{P}$ such that $w^\top x^* = \min_{x \in \mathcal{P}} w^\top x$.

*Problem* 2 (separation problem, SP). Given a vector $y \in \mathbb{R}^d$, decide whether $y$ belong to $\mathcal{P}$ or not, and, in the latter case, find a vector $w \in \mathbb{R}^d$ such that $w^\top y < \min_{x \in \mathcal{P}} w^\top x$.

*Problem* 3 (decomposition problem, DP). Given a vector $x \in \mathcal{P}$, find vertices $x_0, \ldots, x_d$ of $\mathcal{P}$ and $\lambda_0, \ldots, \lambda_d \geq 0$ such that $x = \lambda_0 x_0 + \cdots + \lambda_d x_d$.

Ellipsoid methods provide reductions among these problems, which imply that

$$\text{LP: solvable} \iff \text{SP: solvable} \implies \text{DP: solvable}.$$

**Theorem 3** (**Corollaries 14.1a**, **14.1b** and **14.1g** in [30]). *Suppose that $\mathcal{P} \subseteq \mathbb{R}^d$ is a polytope of which each vertex can be expressed by rationals with bit-lengths of at most $\varphi$, and that each entry of $x, y, w \in \mathbb{Q}^d$ is also a rational, with bit-length of at most $\varphi$. Then, the following holds:*

**(a)** *If there is an algorithm SEP that solves the separation problem, we can solve the linear optimization problem for $w \in \mathbb{Q}^d$ by calling SEP at most $\mathrm{poly}(d, \varphi)$ times.*

**(b)** *If there is an algorithm OPT that solves the linear optimization problem, we can solve the separation problem for $y \in \mathbb{Q}^d$ by calling OPT at most $\mathrm{poly}(d, \varphi)$ times.*

**(c)** *If there is an algorithm OPT that solves the linear optimization problem, we can solve the decomposition problem for $x \in \mathcal{P}$ by calling OPT at most $\mathrm{poly}(d, \varphi)$ times.*

*Remark* 1. For any $\varepsilon > 0$ and any real number $x \in [-1, 1]$, we can approximate $x$ by a rational $\hat{x} \in \mathbb{Q}$ with a bit-length of at most $O(\log(1/\varepsilon))$ so that $|x - \hat{x}| \le \varepsilon$. Hence, we can assume that $\varphi$ in Theorem 3 is bounded as $\varphi = O(\log T)$ by ignoring $O(1/T)$ errors. This implies that the above reductions can be computed in $O(\mathrm{poly}(d, \log T))$ time.

### 4.2 Algorithms for Logconcave Distributions

If a probability distribution over convex body $\mathcal{P} \subseteq \mathbb{R}^d$ has a probability density function (PDF) $p : \mathcal{P} \to \mathbb{R}_{>0}$ such that $\log p$ is a concave function, we refer to it as a *logconcave distribution*. The following theorem means that, given the value oracle of a convex function $f : \mathcal{P} \to \mathbb{R}$, we can approximately sample a vector in $\mathcal{P}$ from a logconcave distribution $p(x) \propto \exp(-f(x))$.

**Theorem 4** (Theorems 2.1 and 2.2 in [25], Lemma 10 in [19]). *Let $\mathcal{P} \subseteq \mathbb{R}^d$ be a convex body with non-zero Lubesgue measure, and let $f : \mathcal{P} \to \mathbb{R}$ be a convex function and let $p$ be a logconcave distribution proportional to $\exp(-f(x))$. Suppose $\varepsilon > 0$ and $\delta \in (0, 1)$. Then, given access to the membership oracle of $\mathcal{P}$ and the value oracle of $f$, there is an algorithm that samples approximately from $p$ such that (i) the total variation distance between the produced distribution and $p$ is at most $\varepsilon$, and (ii) after preprocessing for a time of $O(d^5 (\log d)^{O(1)})$, each sample can be produced in a time of $O(d^4/\varepsilon^4 \cdot (\log(d/\varepsilon))^{O(1)})$.*

As an implication of this theorem, we can efficiently approximate mean $\mu(p) \in \mathbb{R}^d$ and covariance matrix $\mathrm{Cov}(p) \in \mathbb{R}^{d \times d}$ of distribution $p$. In fact, from Corollary 5.52 in [32] and standard concentration of logcancave distribution (see, e.g., Lemma 5.17 in [25]), it takes $(n \log(1/\delta)/\varepsilon)^{O(1)}$ samples to get a matrix $\hat{\Sigma}$ such that $(1 - \varepsilon)\mathrm{Cov}(p) \preceq \hat{\Sigma} \preceq (1 + \varepsilon)\mathrm{Cov}(p)$ with probability of at least $1 - \delta$.[3] Similarly, we can get $\hat{\mu} \in \mathbb{R}^d$ such that $\|\hat{\mu} - \mu(p)\|_{\mathrm{Cov}(p)^{-1}} \le \varepsilon$ from $(n \log(1/\delta)/\varepsilon)^{O(1)}$ samples.[4] Accordingly, we obtain the following corollary:

**Corollary 1.** *Suppose the same assumption as in Theorem 4. There is an algorithm that outputs a vector $\hat{\mu} \in \mathbb{R}^d$ and a symmetric matrix $\hat{\Sigma} \in \mathbb{R}^{d \times d}$ satisfying $\frac{1}{2}\mathrm{Cov}(p) \preceq \hat{\Sigma} \preceq 2\mathrm{Cov}(p)$ and $\|\hat{\mu} - \mu(p)\|_{\mathrm{Cov}(p)^{-1}} \le \varepsilon$ with a probability of at least $1 - \delta$. The computational time, the number of calls for the membership oracle of $\mathcal{P}$, and the value oracle of $f$ are bounded by $\mathrm{poly}(d, \frac{1}{\varepsilon}, \log \frac{1}{\delta})$.*

## 5 Algorithm for Non-stochastic Bandit Linear Optimization

Our algorithm uses the framework of a continuous multiplicative weight update (CMWU) [5; 14; 33]. A straightforward way of applying CMWU is to maintain probability distributions over $\mathcal{K} := \mathrm{Conv}(\mathcal{A})$, which, however, requires a large number of oracle calls. In fact, the algorithm by Hazan and Karnin [19] for bandit linear optimization over convex bodies calls an oracle $\tilde{O}(T^7)$ times. This inefficiency is due to that we need to sample from $\mathcal{K}$; the sampling algorithm in Theorem 4 requires $O(d^4/\varepsilon^4)$-oracle complexity.

We reduce oracle complexity by means of a *cutting-plane approach* [20]. In this approach, we maintain convex bodies $\mathcal{K}_t^{(j)}$ that include and approximate $\mathcal{K}$, and we update a distribution over $\mathcal{K}_t^{(j)}$ instead of $\mathcal{K}$. The advantage of this approach is that we can sample from $\mathcal{K}_t^{(j)}$ without calling

an oracle. On the other hand, updating $\mathcal{K}_t^{(j)}$ requires oracle calls; therefore, we need to bound the number of the updates as well as the number of oracle calls in each update. We design a strategy achieving these as follows: We set *candidates of actions* $\mathcal{E}_t^{(j)} \subseteq \mathcal{K}_t^{(j)}$, from which we choose action. When some actions among the candidates are invalid, i.e., outside of $\mathcal{K}$, we then reduce $\mathcal{K}_t^{(j)}$ by a cutting plane excluding such an invalid candidate. With this strategy, we need oracle calls to check if invalid candidates exist. Our algorithm bounds the oracle complexity here by setting $\mathcal{E}_t^{(j)}$ to have $O(d)$ elements. Further, we design $\mathcal{E}_t^{(j)}$ so that its elements are sufficiently close to the weighted center of $\mathcal{K}_t^{(j)}$. This plays an important role in bounding the number of updates of $\mathcal{K}_t^{(j)}$. Indeed, when a convex body is updated by a cutting plane that excludes a point close to its center, its volume then decreases by a constant factor less than 1 (see, e.g., [25]). On the other hand, $\mathcal{K}_t^{(j)}$ always includes $\mathcal{K}$ with a positive volume; hence, the volume of $\mathcal{K}_t^{(j)}$ cannot be smaller than that of $\mathcal{K}$, which implies that the number of updates is bounded.

## 5.1 Algorithm

Our algorithm maintains a convex body $\mathcal{K}_t^{(j)} \subseteq \mathbb{R}^d$ such that $\mathcal{K}_1^{(0)} \supseteq \mathcal{K}_1^{(1)} \supseteq \cdots \supseteq \mathcal{K}_1^{(s_1)} = \mathcal{K}_2^{(0)} \supseteq \mathcal{K}_2^{(1)} \supseteq \cdots \supseteq \mathcal{K}_2^{(s_2)} \supseteq \cdots \supseteq \mathcal{K}_T^{(s_T)} \supseteq \mathcal{K} = \text{Conv}(\mathcal{A})$, where $t$ corresponds to the round, $j \in \{0, 1, \dots\}$ is an index, and $s_t \in \{0, 1, \dots, T\}$ will be defined later. It also updates a logconcave function $z_t : \mathbb{R}^d \to \mathbb{R}_{>0}$ in each round $t$ based on the multiplicative weight update [5; 14; 33]. Before the first round, $z_t$ is initialized to be a constant function $z_1(x) = 1$. Let $q_t^{(j)}$ denote the PDF of a distribution over $\mathcal{K}_t^{(j)}$ that is proportional to the function $z_t$, i.e.,

$$Z_t^{(j)} = \int_{\mathcal{K}_t^{(j)}} z_t(x)\mathrm{d}x, \quad q_t^{(j)}(x) = \begin{cases} \frac{z_t(x)}{Z_t^{(j)}} & \text{if } a \in \mathcal{K}_t^{(j)}, \\ 0 & \text{if } a \in \mathbb{R}^d \setminus \mathcal{K}_t^{(j)}. \end{cases} \tag{1}$$

Let us denote the mean and the covariance matrix for distribution of $q_t^{(j)}$ by $\mu_t^{(j)} \in \mathbb{R}^d$ and $\Sigma_t^{(j)} \in \mathbb{R}^{d \times d}$, respectively: $\mu_t^{(j)} = \mathbf{E}_{a \sim q_t^{(j)}}[a]$, $\Sigma_t^{(j)} = \mathbf{E}_{a \sim q_t^{(j)}}[(a - \mu_t^{(j)})(a - \mu_t^{(j)})^\top]$. From Corollary 1, we can compute estimators $\hat{\mu}_t^{(j)}$ and $\hat{\Sigma}_t^{(j)}$ of $\mu_t^{(j)}$ and $\Sigma_t^{(j)}$, respectively, such that

$$\frac{1}{2}\Sigma_t^{(j)} \preceq \hat{\Sigma}_t^{(j)} \preceq 2\Sigma_t^{(j)}, \quad \|\hat{\mu}_t^{(j)} - \mu_t^{(j)}\|_{(\Sigma_t^{(j)})^{-1}} \leq \varepsilon \tag{2}$$

with probability of at least $1 - \delta_t^{(j)}$, where $\varepsilon > 0$ and $\delta_t^{(j)} \in (0, 1)$, which will be defined later. Let $B_t^{(j)} = (b_{t1}^{(j)}, \dots, b_{td}^{(j)}) \in \mathbb{R}^{d \times d}$ be a matrix such that $B_t^{(j)} B_t^{(j)\top} = \hat{\Sigma}_d^{(j)}$. Define $\mathcal{E}_t^{(j)} \subseteq \mathbb{R}^d$ as

$$\mathcal{E}_t^{(j)} = \left\{ \hat{\mu}_t^{(j)} + \frac{1}{4\mathrm{e}} b_{ti}^{(j)} \,\middle|\, i \in [d] \right\} \cup \left\{ \hat{\mu}_t^{(j)} - \frac{1}{4\mathrm{e}} b_{ti}^{(j)} \,\middle|\, i \in [d] \right\}. \tag{3}$$

In each round $t$, our algorithm checks if $\mathcal{E}_t^{(j)}$ is included in $\mathcal{K}$, and if not, it updates $\mathcal{K}_t^{(j)}$, as described in Step 7 of Algorithm 1, to exclude an element in $\mathcal{E}_t^{(j)} \setminus \mathcal{K}$. Set $\mathcal{E}_t^{(j)}$ is designed so that the following four conditions are satisfied:

1. The cardinality of $\mathcal{E}_t^{(j)}$ is bounded as $|\mathcal{E}_t^{(j)}| = O(d)$. Hence, we can decide if $\mathcal{E}_t^{(j)} \subseteq \mathcal{K}$ by $O(\text{poly}(d))$ oracle calls.

2. Each $y \in \mathcal{E}_t^{(j)}$ is sufficiently close to $\mu_t^{(j)}$, i.e., it satisfies $\|y - \mu_t^{(j)}\|_{(\Sigma_t^{(j)})^{-1}} \leq 1/(2\mathrm{e})$. This is important to bound the number of oracle calls.

3. The mean of $\mathcal{E}_t^{(j)}$ is equal to $\hat{\mu}_t^{(j)}$. This implies that if $y$ follows a uniform distribution over $\mathcal{E}_t^{(j)}$, we then have $\mathbf{E}[\ell_t^\top y] = \ell_t^\top \hat{\mu}_t^{(j)} \approx \ell_t^\top \mu_t^{(j)} = \mathbf{E}[\ell_t^\top x]$ for $x \sim q_t^{(j)}$.

4. The covariance matrix $\Sigma$ of a uniform distribution over $\mathcal{E}_t^{(j)}$ satisfies $\Sigma \succeq O(1/d^2) \cdot \Sigma_t^{(j)}$. Thanks to this, empirical estimates of $\ell_t$ based on $\mathcal{E}_t^{(j)}$ will have a sufficiently small variance.

The conditions 1 and 2 are used to bound the oracle complexity, and 3 and 4 are necessary to bound the regret. Once $\mathcal{E}_t^{(j)}$ is included in $\mathcal{K}$, our algorithm escapes the loop of updating $\mathcal{K}_t^{(j)}$. An integer $s_t$

**Algorithm 1** An oracle efficient algorithm for non-stochastic bandit linear optimization

---

**Require:** Learning rate $\eta > 0$, error bound $\varepsilon > 0$, time horizon $T \in \mathbb{N}$, $R > 0$ satisfying Assumption 1.

1: Set $\mathcal{K}_1^{(0)} = B_\infty(0, R) = \{x \in \mathbb{R}^d \mid \|x\|_\infty \leq R\}$ and define $z_1 : \mathbb{R}^d \to \mathbb{R}_{>0}$ by $z_1(x) = 1$.
2: **for** $t = 1, 2, \ldots, T$ **do**
3:     **for** $j = 0, 1, 2, \ldots$ **do**
4:         Compute $\mathcal{E}_t^{(j)}$ on the basis of (1) $\sim$ (3).
5:         Solve SP for $\mathcal{P} = \mathcal{K}$ and for each $y \in \mathcal{E}_t^{(j)}$.
6:         **if** There is a hyperplane $w \in \mathbb{R}^d$ s.t. $w^\top y < \min_{x \in \mathcal{K}} w^\top x$ for some $y \in \mathcal{E}_t^{(j)}$ **then**
7:             Update $\mathcal{K}_t^{(j)}$ by $\mathcal{K}_t^{(j+1)} = \mathcal{K}_t^{(j)} \cap \{x \in \mathbb{R}^d \mid w^\top x \geq w^\top y\}$.
8:         **else**
9:             Set $s_t = j$ and $\mathcal{K}_t = \mathcal{K}_t^{(s_t)}$. Break the **for** loop w.r.t. the index $j$.
10:         **end if**
11:     **end for**
12:     Let $\hat{\mu}_t = \hat{\mu}_t^{(s_t)}$ and $b_{ti} = b_{ti}^{(s_t)}$ for $i \in [n]$, which are defined in (1) $-$ (3).
13:     Choose $i_t \in [d]$ and $\sigma_t \in \{1, -1\}$ uniformly at random.
14:     Solve DP for $\mathcal{P} = \mathcal{K}$ and $x = \hat{\mu}_t + \frac{\sigma_t}{4e} b_{ti_t}$ to get a decomposition $x_{t0}, \ldots, x_{td} \in \mathcal{A}$ and $\lambda_{t0}, \ldots, \lambda_{td}$ such that $\hat{\mu}_t + \frac{\sigma_t}{4e} b_{ti_t} = \lambda_{t0} x_{t0} + \cdots + \lambda_{td} x_{td}$.
15:     Play $a_t = x_{ts}$ with probability $\lambda_{ts}$ ($s = 0, \ldots, d$), and receive loss $\ell_t^\top a_t$.
16:     Set $\hat{\ell}_t$ by (4) and update $z_t$ by $z_{t+1}(x) = z_t(x) \exp(-\eta \hat{\ell}_t^\top (x - \hat{\mu}_t))$.
17:     Set $\mathcal{K}_{t+1}^{(0)} = \mathcal{K}_t^{(s_t)}$.
18: **end for**

---

denotes the number of the updates in the round $t$. We denote $\mathcal{E}_t = \mathcal{E}_t^{(s_t)}$, $\hat{\Sigma}_t = \hat{\Sigma}_t^{(s_t)}$, $\hat{\mu}_t = \hat{\mu}_t^{(s_t)}$, and $B_t = B_t^{(s_t)}$. We randomly choose $x$ from $\mathcal{E}_t$ as follows: choose $\sigma_t \in \{-1, 1\}$ and $i_t \in [d]$ uniformly at random, and define $x = \hat{\mu}_t + \frac{\sigma_t}{4e} b_{ti_t}$. If we play this $x$, then we can construct a good estimate of $\ell_t$ from the above condition 4, which leads to a small degree of regret. However, $x \in \mathcal{E}_t$ does not always belong to $\mathcal{A}$, particularly when $\mathcal{A}$ is discrete. To address this issue, we solve DP for this $x$ and $\mathcal{P} = \mathcal{K}$ to derive a decomposition of $x$, i.e., compute $x_{t0}, \ldots, x_{td} \in \mathcal{K}$ and $\lambda_{t0}, \ldots, \lambda_{td} \geq 0$ as in Step 14. Then, the algorithm plays $a_t = x_{ti}$ with probability $\lambda_{ti}$, and obtains feedback of $\ell_t^\top a_t$. Based on this feedback, we compute an estimator $\hat{\ell}_t$ of the loss vector $\ell_t$ as

$$\hat{\ell}_t = 4ed\sigma_t \ell_t^\top a_t \hat{\Sigma}_t^{-1} b_{ti_t}. \tag{4}$$

This is an unbiased estimator of $\ell_t$, i.e., we have $\mathbf{E}[\hat{\ell}_t] = \ell_t$. The existence of $\hat{\Sigma}_t^{-1}$ follows from the definition of $\hat{\Sigma}$ and Assumption 1. In fact, from $\mathcal{A} \subseteq \mathcal{K}_t^{(j)}$ and Assumption 1, $\mathcal{K}_t^{(j)}$ has a positive volume and $q_t^{(j)}$ has a positive density over $\mathcal{K}_t^{(j)}$, which implies that the covariance matrix $\Sigma_t^{(j)}$ of $q_t^{(j)}$ is positive definite. From this and (2), $\hat{\Sigma}_t^{(j)}$ is positive definite for all $t$ and $j$. The function $z_t$ is updated by $z_{t+1}(x) = z_t(x) \exp(-\eta \hat{\ell}_t^\top (x - \hat{\mu}_t))$, where $\eta > 0$ is an input parameter standing for the learning rate, which will be optimized later. Let

$$\delta_t^{(j)} = 1/(T(j + 2 + \sum_{i=1}^{t-1}(s_i + 1))(j + 3 + \sum_{i=1}^{t-1}(s_i + 1))). \tag{5}$$

To compute $\hat{\Sigma}_t^{(j)}$ and $\hat{\mu}_t^{(j)}$ satisfying (2) with probability at least $1 - \delta_t^{(j)}$, we use the algorithm in Corollary 1.

Let $S_T = \sum_{t=1}^T s_t$ denote the number of updates of $\mathcal{K}_t^{(j)}$. We show the following regret bound.

**Theorem 5.** *Define $\psi = \frac{1}{d} \log \frac{\text{Vol}(B_\infty(0,R))}{\text{Vol}(\mathcal{K})}$. Suppose $a_t$ is given by Algorithm 1 with parameters $\varepsilon = \frac{1}{12eT}$ and $\eta = \frac{1}{2eLR} \min\{\sqrt{\frac{1+\psi+\log T}{dT}}, \frac{1}{2^4 d^{3/2}(1+\psi+\log T)}\}$. Then, for all $a^* \in \mathcal{A}$, we have*

$$\mathbf{E}[R_T(a^*)] \leq 2^7 eLRd^{3/2} \max\{\sqrt{T(1 + \psi + \log T)}, d(1 + \psi + \log T)^2\} \left(1 - S_T/2^{10}\right). \tag{6}$$

We note that $\psi$ in the above theorem satisfies $\psi \leq \log \frac{R}{r}$ if $\mathcal{K}$ includes an $\ell_\infty$-ball of radius $r > 0$. The proof of this theorem is given in Appendix A.

## 5.2 Oracle Complexity Analysis

Here, we show that Algorithm 1 calls the linear optimization oracles only $O(\text{poly}(d)T)$ times.

To implement Algorithm 1, the linear optimization oracle is required only in Steps 5 and 14. In Step 5, we need to solve SP to decide if there exists $x \in \mathcal{E}_t^{(j)}$ such that $x \notin \mathcal{K}$. From the definition (3) of $\mathcal{E}_t^{(j)}$, the number of elements in $\mathcal{E}_t^{(j)}$ is equal to $2d$ for each $t$ and $j$, and, accordingly, the total number of solutions of SP is $\sum_{t=1}^{T} \sum_{j=0}^{s_t} |\mathcal{E}_t^{(j)}| = 2d \sum_{t=1}^{T} (s_t + 1) = 2d(T + S_T)$. The number $S_T$ can be bounded as $S_T = O(T)$. Indeed, from Theorem 5, if $S_T > 2^{10}(1+T)$ then $\mathbf{E}[R_T(a^*)] < -2^7 \mathrm{e} LRT$, which contradicts to $R_T(a^*) = \sum_{t=1}^{T} \ell_t^\top (a_t - a^*) \geq -2LRT$. Consequently, the total number of solutions of SP is $O(dT)$. In Step 14, we solve DP in each round $t$; hence the total number of solutions of DP is equal to $T$. Because we can solve SP and DP by calling the linear optimization oracle $\text{poly}(d, \log T)$ times from Theorem 4 and Remark 1, we can implement Algorithm 1 so that it calls the linear optimization oracle $O(\text{poly}(d, \log T)T)$ times.

## Footnotes

[1] In $\tilde{O}(\cdot)$ notation, we ignore factors of polynomials in $d$ and $\log(T)$.

[2] In this algorithm, $\mathcal{A}$ is assumed to be a convex body, and a membership oracle for $\mathcal{A}$ is assumed. Because we can construct a membership oracle from a linear optimization oracle and vice versa by a polynomial-time reduction [30], the assumption regarding the oracle is equivalent to ours, modulo polynomial-time reduction.

[3] A similar argument can be found in Section 6.3 in [10].

[4] For a vector $x \in \mathbb{R}^d$ and a positive semidefinite matrix $A \in \mathbb{R}^{d \times d}$, denote $\|x\|_A := \sqrt{x^\top A x}$.

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
