[Supplementary Material · main.pdf]

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

## A  Regret Analysis of Algorithm 1

We use the following two lemmas to prove Theorem 5

**Lemma 1.** *The conditional expectation of $\hat{\ell}_t$ defined by* (4)*, given $\ell_t$ and $\mathcal{E}_t$, satisfies*

$$\mathbf{E}[\hat{\ell}_t] = \ell_t.$$

*Proof.* Because $\sum_{s=0}^d \lambda_{ts} x_{ts} = \hat{\mu}_t + \frac{\sigma_t}{4\mathrm{e}} b_{ti_t}$, the expectation of $a_t$ given $\sigma_t, i_t$ satisfies

$$\mathbf{E}[a_t] = \hat{\mu}_t + \frac{\sigma_t}{4\mathrm{e}} b_{ti_t}.$$

Hence, we have

$$\mathbf{E}[\sigma_t b_{ti_t} a_t^\top] = \mathbf{E}\left[\sigma_t b_{ti_t}\left(\hat{\mu}_t + \frac{\sigma_t}{4\mathrm{e}} b_{ti_t}\right)^\top\right] = \frac{1}{4\mathrm{e}}\mathbf{E}[b_{ti_t} b_{ti_t}^\top] = \frac{1}{4\mathrm{e}d}\sum_{i=1}^d b_{ti} b_{ti}^\top = \frac{1}{4\mathrm{e}d}\hat{\Sigma}_t,$$

where the second equality comes from $\mathbf{E}[\sigma_t] = 0$ and $\sigma_t^2 = 1$, the third equality holds because $i_t$ follows a uniform distribution over $[d]$, and the last equality follows from the definition of $\{b_{ti}\}_{i=1}^d$. From the above equation and definition (4) of $\hat{\ell}_t$, we have

$$\mathbf{E}[\hat{\ell}_t] = 4\mathrm{e}d\hat{\Sigma}_t^{-1}\mathbf{E}[\sigma_t b_{ti_t} a_t^\top]\ell_t = \ell_t.$$

$\square$

**Lemma 2.** *Suppose that a random variable $X$ follows a logconcave distribution and that $\mathbf{E}[X^2] \leq 1/\alpha^2$ holds for given $\alpha \geq 1$. Then, we have*

$$\mathbf{E}[\exp(X)\mathbf{1}_{X>1}] \leq \frac{\exp(3-\alpha)}{1-\exp(1-\alpha)}.$$

*Proof.* From Lemma 5.7 in [25], we have

$$\mathrm{Prob}[|X| \geq i] \leq \exp(-\alpha i + 1)$$

for all $i \geq 1$. Hence, we have

$$\mathbf{E}[\exp(X)\mathbf{1}_{X>1}] = \sum_{i=1}^\infty \mathbf{E}[\exp(X)\mathbf{1}_{i \leq X \leq i+1}] \leq \sum_{i=1}^\infty \mathrm{Prob}[i \leq X \leq i+1]\exp(i+1)$$

$$\leq \sum_{i=1}^\infty \mathrm{Prob}[|X| \geq i]\exp(i+1) \leq \sum_{i=1}^\infty \exp((1-\alpha)i + 2) = \frac{\exp(3-\alpha)}{1-\exp(1-\alpha)}.$$

$\square$

To prove Theorem 5, we introduce some notations. In the following, we denote

$$f_t(x) = \ell_t^\top(x - \hat{\mu}_t), \quad \hat{f}_t(x) = \hat{\ell}_t^\top(x - \hat{\mu}_t). \tag{7}$$

Then, we can express $z_t$ as

$$z_t(x) = \exp\left(-\eta \sum_{i=1}^{t-1} \hat{f}_i(x)\right). \tag{8}$$

Then, the regret can be bounded by means of $Z_{T+1}^{(0)}$ defined in (1), as follows:

**Lemma 3.** *For all $a^* \in \mathcal{A}$ and $\gamma \in (0,1)$, we have*

$$\mathbf{E}[R_T(a^*)] \leq \frac{1}{\eta(1-\gamma)}\left(\mathbf{E}\log Z_{T+1}^{(0)} - \log \mathrm{Vol}(\mathcal{K}) + \eta\gamma LRT - d\log\gamma\right). \tag{9}$$

*Proof.* From $\mathbf{E}[a_t|\hat{\mu}_t] = \hat{\mu}_t$ and $\mathbf{E}[\hat{\ell}_t|\hat{\mu}_t] = \ell_t$, we have

$$\mathbf{E}\,R_T(a^*) = \mathbf{E}\sum_{t=1}^{T}\ell_t^\top(a_t - a^*) = \mathbf{E}\sum_{t=1}^{T}\ell_t^\top(\hat{\mu}_t - a^*) = \mathbf{E}\sum_{t=1}^{T}\hat{\ell}_t^\top(\hat{\mu}_t - a^*) = -\mathbf{E}\sum_{t=1}^{T}\hat{f}_t(a^*).$$
(10)

We consider evaluating the rightmost-hand side, by using $Z_{T+1}^{(0)}$. Define a convex body $\bar{\mathcal{K}}$ by $\bar{\mathcal{K}} := (1-\gamma)a^* + \gamma\mathcal{K}$. Because $\mathcal{K}$ is convex and $a^* \in \mathcal{K}$, we have $\bar{\mathcal{K}} \subseteq \mathcal{K}$. Hence, we have $\bar{\mathcal{K}} \subseteq \mathcal{K}_t^{(j)}$ for all $t$ and $j$. Then, we have

$$\begin{aligned}
\mathbf{E}\log Z_{T+1}^{(0)} &= \mathbf{E}\log\int_{\mathcal{K}_T} z_{T+1}(x)\mathrm{d}x \geq \mathbf{E}\log\int_{\bar{\mathcal{K}}} z_{T+1}(x)\mathrm{d}x \\
&= \mathbf{E}\log\int_{\mathcal{K}}\gamma^d z_{T+1}((1-\gamma)a^* + \gamma x)\mathrm{d}x \\
&= d\log\gamma + \mathbf{E}\log\int_{\mathcal{K}}\exp\left(-\eta\sum_{t=1}^{T}((1-\gamma)\hat{f}_t(a^*) + \gamma\hat{f}_t(x))\right)\mathrm{d}x \\
&= d\log\gamma - \eta(1-\gamma)\mathbf{E}\sum_{t=1}^{T}\hat{f}_t(a^*) + \mathbf{E}\log\int_{\mathcal{K}}\exp(-\eta\gamma\sum_{t=1}^{T}\hat{f}_t(x))\mathrm{d}x.
\end{aligned}$$
(11)

The factor $\gamma^d$ in the second equality comes from the change of variables $x \leftarrow (1-\gamma)a^* + \gamma x$. The last term in (11) can be bounded from below by means of $\bar{x} := \frac{\int_{\mathcal{K}} x\mathrm{d}x}{\int_{\mathcal{K}} 1\mathrm{d}x} \in \mathcal{K}$. Indeed, because $\hat{f}_t$ is an affine function and $\exp$ is a convex function, from Jensen's inequality, we have

$$\mathbf{E}\log\int_{\mathcal{K}}\exp(-\eta\gamma\sum_{t=1}^{T}\hat{f}_t(x))\mathrm{d}x \geq \mathbf{E}\log\int_{\mathcal{K}}\exp(-\eta\gamma\sum_{t=1}^{T}\hat{f}_t(\bar{x}))\mathrm{d}x$$

$$= -\eta\gamma\,\mathbf{E}\sum_{t=1}^{T}\hat{f}_t(\bar{x}) + \log\int_{\mathcal{K}}1\mathrm{d}x = -\eta\gamma\,\mathbf{E}\sum_{t=1}^{T}f_t(\bar{x}) + \log\int_{\mathcal{K}}1\mathrm{d}x \geq -\eta\gamma LRT + \log\mathrm{Vol}(\mathcal{K}).$$

By combining this and (11), we obtain

$$-\eta(1-\gamma)\mathbf{E}\sum_{t=1}^{T}\hat{f}_t(a^*) \leq \mathbf{E}\log Z_{T+1}^{(0)} - \log\mathrm{Vol}(\mathcal{K}) + \eta\gamma LRT - d\log\gamma.$$

From this and (10), we have (9). □

The value $\mathbf{E}\log Z_{T+1}^{(0)}$ can be expressed as

$$\mathbf{E}\log Z_{T+1}^{(0)} = \log Z_1^{(0)} + \sum_{t=1}^{T}\mathbf{E}\log\frac{Z_{t+1}^{(0)}}{Z_t^{(0)}} \leq \log Z_1^{(0)} + \sum_{t=1}^{T}\left(\mathbf{E}\log\frac{Z_{t+1}^{(0)}}{Z_t^{(s_t)}} + \mathbf{E}\log\frac{Z_t^{(s_t)}}{Z_t^{(0)}}\right).$$
(12)

We can evaluate $\mathbf{E}\log\frac{Z_{t+1}^{(0)}}{Z_t^{(s_t)}}$ and $\mathbf{E}\log\frac{Z_t^{(s_t)}}{Z_t^{(0)}}$ as in Lemmas 4 and 5, respectively.

**Lemma 4.** *Under the assumption that* (2) *and* $\eta \leq \frac{1}{2^3 \mathrm{e} LRd^{3/2}(4+\log T)}$ *holds for all* $t \in [T]$, *we have*

$$\mathbf{E}\log\frac{Z_{t+1}^{(0)}}{Z_t^{(s_t)}} \leq LR\varepsilon\eta + 2^5(\mathrm{e}dLR\eta)^2 + \frac{1}{T}.$$
(13)

*Proof.* We denote $q_t := q_t^{(s_t)}$, which is defined in (1). Let $\mu_t \in \mathbb{R}^d$ and $\Sigma_t \in \mathbb{R}^{d\times d}$ denote the mean and the covariance matrix of $q_t$, respectively. Define $\tilde{f}_t : \mathbb{R}^d \to \mathbb{R}$ by $\tilde{f}_t(x) = \hat{\ell}_t^\top(x - \mu_t)$. From

definition (7) of $\hat{f}_t$, we have $\hat{f}_t(x) = \tilde{f}_t(x) + \hat{\ell}_t^\top(\mu_t - \hat{\mu}_t)$. From definitions (8) and (1) of $z_t$ and $Z_t^j$, we can express $\frac{Z_{t+1}^{(0)}}{Z_t^{(s_t)}}$ as follows:

$$\frac{Z_{t+1}^{(0)}}{Z_t^{(s_t)}} = \int_{\mathcal{K}_t} \frac{z_{t+1}(x)}{Z_t^{(s_t)}} \mathrm{d}x = \int_{\mathcal{K}_t} \frac{z_t(x)}{Z_t^{(s_t)}} \exp(-\eta \hat{f}_t(x)) \mathrm{d}x$$

$$= \mathop{\mathbf{E}}_{x \sim q_t} \exp(-\eta \hat{f}_t(x)) = \exp(-\eta \hat{\ell}_t^\top (\mu_t - \hat{\mu}_t)) \cdot \mathop{\mathbf{E}}_{x \sim q_t} \exp(-\eta \tilde{f}_t(x)). \qquad (14)$$

Because $\exp(x) \leq 1 + x + x^2$ holds for $x \leq 1$, $\mathop{\mathbf{E}}_{x \sim q_t} \exp(-\eta \tilde{f}_t(x))$ can be bounded as

$$\mathop{\mathbf{E}}_{x \sim q_t} \exp(-\eta \tilde{f}_t(x)) \leq \mathop{\mathbf{E}}_{x \sim q_t} [(1 - \eta \tilde{f}_t(x) + \eta^2 \tilde{f}_t(x)^2)\mathbf{1}_{-\eta \tilde{f}_t(x) \leq 1}] + \mathop{\mathbf{E}}_{x \sim q_t} [\exp(-\eta \tilde{f}_t(x))\mathbf{1}_{-\eta \tilde{f}_t(x) > 1}]$$

$$\leq \mathop{\mathbf{E}}_{x \sim q_t} [(1 - \eta \tilde{f}_t(x) + \eta^2 \tilde{f}_t(x)^2)] + \mathop{\mathbf{E}}_{x \sim q_t} [\exp(-\eta \tilde{f}_t(x))\mathbf{1}_{-\eta \tilde{f}_t(x) > 1}]. \qquad (15)$$

The first term in (15) can be, from definition of $\mu_t$, $\Sigma_t$, and $\tilde{f}_t$, expressed as follows:

$$\mathop{\mathbf{E}}_{x \sim q_t} [(1 - \eta \tilde{f}_t(x) + \eta^2 \tilde{f}_t(x)^2)] = 1 - \eta \mathop{\mathbf{E}}_{x \sim q_t} [\hat{\ell}_t^\top (x - \mu_t)] + \eta^2 \mathop{\mathbf{E}}_{x \sim q_t} [\hat{\ell}_t^\top (x - \mu_t)(x - \mu_t)\hat{\ell}_t^\top]$$

$$= 1 + \eta^2 \hat{\ell}_t^\top \Sigma_t \hat{\ell}_t. \qquad (16)$$

To bound the second term in (15), we use Lemma 2 with $X = -\eta \tilde{f}_t(x)$: If $x$ follows $q_t$, a log-concave distribution, and if $\hat{\ell}_t$ is fixed, i.e., $\ell_t$, $\sigma_t$, $i_t$ and $a_t$ are fixed, then $\tilde{f}_t(x) = \hat{\ell}_t(x - \mu_t) = 4\mathrm{e}d\sigma_t \ell_t^\top a_t b_{t i_t} \hat{\Sigma}_t^{-1}(x - \mu_t)$ follows a logconcave distribution, because logconcavity is preserved under linear transformations (see, e.g., [28]). Furthermore, we have[5]

$$\mathop{\mathbf{E}}_{x \sim q_t} [(\eta \tilde{f}_t(x))^2] = \eta^2 \mathop{\mathbf{E}}_{x \sim q_t} [\hat{\ell}_t^\top (x - \mu_t)(x - \mu_t)^\top \hat{\ell}_t] = \eta^2 \hat{\ell}_t^\top \Sigma_t \hat{\ell}_t = (4\mathrm{e}d\eta \ell_t^\top a_t)^2 b_{t i_t}^\top \hat{\Sigma}_t^{-1} \Sigma_t \hat{\Sigma}_t^{-1} b_{t i_t}$$

$$\leq 2(4\mathrm{e}d\eta LR)^2 b_{t i_t}^\top \hat{\Sigma}_t^{-1} b_{t i_t} = 2(4\mathrm{e}d\eta LR)^2 \hat{\Sigma}_t^{-1} \bullet (b_{t i_t} b_{t i_t}^\top) \leq 2^5 (\mathrm{e}d\eta LR)^2 \hat{\Sigma}_t^{-1} \bullet \hat{\Sigma}_t = 2^5 (\mathrm{e}\eta LR)^2 d^3,$$

where the first, second, and third equalities come from the definitions of $\tilde{f}_t$, $\Sigma_t$, and $\hat{\ell}_t$, respectively, the first inequality follows from the condition $\Sigma_t \preceq 2\hat{\Sigma}_t$ given in (2), and the second inequality follows from that $\hat{\Sigma}_t = \sum_{i=1}^d b_{ti} b_{ti}^\top$. Hence, under the assumption that $2^5 (\mathrm{e}\eta LR)^2 d^3 (4 + \log T)^2 \leq 1$ holds, it follows from Lemma 2 that

$$\mathop{\mathbf{E}}_{x \sim q_t} [\exp(-\eta \tilde{f}_t(x))\mathbf{1}_{-\eta \hat{f}_t(x) > 1}] \leq \frac{\exp(-1 - \log T)}{1 - \exp(-3 - \log T)} \leq \frac{1}{T}.$$

Combining this, (15), and (16), we obtain

$$\mathop{\mathbf{E}}_{x \sim q_t} \exp(-\eta \tilde{f}_t(x)) \leq 1 + \eta^2 \hat{\ell}_t^\top \Sigma_t \hat{\ell}_t + \frac{1}{T}.$$

From this, (14), Lemma 1, and the fact that $\log(1 + x) \leq x$ holds for $x \geq 0$, we have

$$\mathbf{E} \log \frac{Z_{t+1}^{(0)}}{Z_t^{(s_t)}} \leq -\eta \mathbf{E} \, \hat{\ell}_t^\top (\mu_t - \hat{\mu}_t) + \mathbf{E} \log \left(1 + \eta^2 \hat{\ell}_t^\top \Sigma_t \hat{\ell}_t + \frac{1}{T}\right)$$

$$\leq -\eta \mathbf{E} \, \ell_t^\top (\mu_t - \hat{\mu}_t) + \mathbf{E} \left[\eta^2 \hat{\ell}_t^\top \Sigma_t \hat{\ell}_t\right] + \frac{1}{T}.$$

From (2), we have

$$|\ell_t^\top (\mu_t - \hat{\mu}_t)| \leq \|\ell_t\|_2 \|\mu_t - \hat{\mu}_t\|_2 \leq L \|\mu_t - \hat{\mu}_t\|_{\Sigma_t^{-1}} \|\Sigma_t\|_2 \leq LR\varepsilon,$$

where $\|\Sigma_t\|_2$ stands for the $\ell_2$ operator norm, i.e., the largest eigenvalue of $\Sigma_t$, and the last inequality holds because $\Sigma_t$ is the covariance matrix of distribution over a region included in $B_\infty(0, R)$. Furthermore, we have

$$\mathbf{E} \left[\hat{\ell}_t^\top \Sigma_t \hat{\ell}_t\right] \leq (4\mathrm{e}dLR)^2 \Sigma_t^{-1} \bullet \mathbf{E}[b_{t i_t} b_{t i_t}^\top] = (4\mathrm{e}dLR)^2 \Sigma_t^{-1} \bullet \left(\frac{1}{d}\sum_{i=1}^d b_{ti} b_{ti}^\top\right)$$

$$= (4\mathrm{e}dLR)^2 \Sigma_t^{-1} \bullet \left(\frac{1}{d}\hat{\Sigma}_t\right) \leq 2(4\mathrm{e}dLR)^2.$$

By combining the above three inequalities, we obtain (13). $\qquad\square$

**Lemma 5.** *Suppose $\varepsilon \leq 1/(12\mathrm{e})$. For all $t \in [T]$ and $j \in \{0, 1, \ldots, s_t - 1\}$, under the assumption of (2), we have*

$$\frac{Z_t^{(j+1)}}{Z_t^{(j)}} \leq \left(1 - \frac{1}{2\mathrm{e}}\right). \tag{17}$$

*Proof.* Let $(w, b)$ denote the hyperplane that the algorithm chooses for updating $\mathcal{K}_t^{(j)}$, which means that $\mathcal{K}_t^{(j+1)} = \mathcal{K}_t^{(j)} \cap \{x \in \mathbb{R}^d \mid w^\top x \geq b\}$. Then, we have

$$1 - \frac{Z_t^{(j+1)}}{Z_t^{(j)}} = \frac{\int_{\mathcal{K}_t^{(j)} \setminus \mathcal{K}_t^{(j+1)}} z_t(a)\mathrm{d}a}{\int_{\mathcal{K}_t^{(j)}} z_t(a)\mathrm{d}a} = \Pr_{x \sim q_t^{(j)}}[w^\top x < b].$$

Because $(w, b)$ satisfies $w^\top x \leq b$ for some $x \in \mathcal{E}_t^{(j)}$, there exists $i \in [d]$ and $\sigma \in \{1, -1\}$ such that

$$w^\top \left(\hat{\mu}_t^{(j)} + \frac{\sigma}{4\mathrm{e}} b_{ti}^{(j)}\right) \leq b.$$

Combining the above equality and inequality, we obtain

$$1 - \frac{Z_t^{(j+1)}}{Z_t^{(j)}} \geq \Pr_{x \sim q_t^{(j)}} \left[w^\top x < w^\top \left(\hat{\mu}_t^{(j)} + \frac{\sigma}{4\mathrm{e}} b_{ti}^{(j)}\right)\right]$$

$$= \Pr_{x \sim q_t^{(j)}} \left[w^\top \left(x - \mu_t^{(j)}\right) < w^\top \left(\hat{\mu}_t^{(j)} - \mu_t^{(j)} + \frac{\sigma}{4\mathrm{e}} b_{ti}^{(j)}\right)\right].$$

The value $w^\top \left(\hat{\mu}_t^{(j)} - \mu_t^{(j)} + \frac{\sigma}{4\mathrm{e}} b_{ti}^{(j)}\right)$ can be bounded as

$$\left| w^\top \left(\hat{\mu}_t^{(j)} - \mu_t^{(j)} + \frac{\sigma}{4\mathrm{e}} b_{ti}^{(j)}\right) \right| \leq \|w\|_{\hat{\Sigma}_t^{(j)}} \left(\|\hat{\mu}_t^{(j)} - \mu_t^{(j)}\|_{\hat{\Sigma}_t^{(j)-1}} + \frac{1}{4\mathrm{e}} \|b_{ti}^{(j)}\|_{\hat{\Sigma}_t^{(j)-1}}\right)$$

$$\leq \|w\|_{\hat{\Sigma}_t^{(j)}} \left(\varepsilon + \frac{1}{4\mathrm{e}}\right) \leq \sqrt{2}\|w\|_{\Sigma_t^{(j)}} \left(\varepsilon + \frac{1}{4\mathrm{e}}\right) \leq \frac{\|w\|_{\Sigma_t^{(j)}}}{2\mathrm{e}},$$

where the first inequality comes from the Cauchy–Schwarz inequality, the second inequality follows from (2) and that $\Sigma_t^{(j)} = \sum_{i=1}^d b_{ti}^{(j)} b_{ti}^{(j)\top}$, the third inequality follows from $\hat{\Sigma}_t^{(j)} \succeq 2\Sigma_t^{(j)}$ in (2), and the last inequality comes from the assumption of $\varepsilon < 1/(12\mathrm{e})$. From the above two inequations, we have

$$1 - \frac{Z_t^{(j+1)}}{Z_t^{(j)}} \geq \Pr_{x \sim q_t^{(j)}} \left[\frac{w^\top(x - \mu_t^{(j)})}{\|w\|_{\Sigma_t^{(j)}}} < -\frac{1}{2\mathrm{e}}\right].$$

When $x$ follows $q_t^{(j)}$, $y := \frac{w^\top(x - \mu_t^{(j)})}{\|w\|_{\Sigma_t^{(j)}}}$ follows a distribution with mean 0 and variance 1, because $w^\top x$ has mean $w^\top \mu_t^{(j)}$ and variance $w^\top \Sigma_t^{(j)} w$. Moreover, the PDF of $y$ is a logconcave function, because logconcavity is preserved under linear transformations [28]. Because we have $\Pr[y \leq 0] \geq 1/\mathrm{e}$ from Lemma 5.4 in [25] and $\Pr[-1/(2\mathrm{e}) \leq y \leq 0] \leq 1/(2\mathrm{e})$ from Lemma 5.5 in [25], we have $\Pr[y < -1/(2\mathrm{e})] \geq 1/(2\mathrm{e})$. $\qquad\square$

Combining (12) and Lemmas 4 and 5, under the assumption that (2) holds for all $t \in [T]$ and $j \in \{0, 1, \ldots, s_t\}$, we have

$$\mathbf{E} \log Z_{T+1}^{(0)} \leq \log Z_1^{(0)} + T\left(LR\varepsilon\eta + 2^5(\mathrm{ed}LR\eta)^2 + \frac{1}{T}\right) + \sum_{t=1}^T s_t \log\left(1 - \frac{1}{2\mathrm{e}}\right)$$

$$\leq \log Z_1^{(0)} + LRT\varepsilon\eta + 2^5 T(\mathrm{ed}LR\eta)^2 + 1 - \frac{S_T}{5}, \tag{18}$$

where we denote $S_T = \sum_{t=1}^T s_t$ and the second inequality follows from $\log\left(1 - \frac{1}{2\mathrm{e}}\right) \leq \frac{1}{5}$. Define $\delta := \Pr[\text{there exists } t \in [T] \text{ and } j \leq s_t \text{ such that (2) does not hold}]$. From definition (5) of $\delta_t^{(j)}$,

we have $\delta \leq \sum_{t=1}^{T} \sum_{j=0}^{s_t} \leq \sum_{k=2}^{\infty} \frac{1}{Tk(k+1)} = \frac{1}{2T}$. Because (18) holds with probability at least $1 - \delta$, and it always holds that $R_T(a^*) \leq 2LRT$, from Lemma 3, we have

$$\mathbf{E}[R_T(a^*)] \leq \frac{1-\delta}{1-\gamma}\left(2^5T(edLR)^2\eta + \frac{1}{\eta}\left(\log\frac{Z_1^{(0)}}{\mathrm{Vol}(\mathcal{K})} + 1 - d\log\gamma - \frac{S_T}{5}\right) + LRT(\varepsilon + \gamma)\right) + 2\delta LRT$$

$$\leq \frac{1}{1-\gamma}\left(2^5T(edLR)^2\eta + \frac{1}{\eta}\left(\log\frac{Z_1^{(0)}}{\mathrm{Vol}(\mathcal{K})} + 1 - d\log\gamma - \frac{S_T}{10}\right) + LRT(\varepsilon + \gamma)\right) + LR,$$

where the second inequality comes from $0 \leq \delta \leq \frac{1}{2T}$. We denote $\psi := \frac{1}{d}\log\frac{Z_1^{(0)}}{\mathrm{Vol}(\mathcal{K})} = \frac{1}{d}\log\frac{\mathrm{Vol}(B_\infty(0,R))}{\mathrm{Vol}(\mathcal{K})}$. By setting $\varepsilon = \gamma = \frac{1}{12eT}$, we obtain

$$\mathbf{E}[R_T(a^*)] \leq 2\left(2^5T(edLR)^2\eta + \frac{1}{\eta}\left(d(5 + \psi + \log T) - \frac{S_T}{10}\right) + 2LR\right).$$

In addition, by setting $\eta = \frac{1}{2eLR}\min\left\{\sqrt{\frac{1+\psi+\log T}{dT}}, \frac{1}{2^4d^{3/2}(1+\psi+\log T)}\right\}$, we obtain

$$\mathbf{E}[R_T(a^*)] \leq 2^7eLRd^{3/2}\max\left\{\sqrt{T(1 + \psi + \log T)}, d(1 + \psi + \log T)^2\right\}\left(1 - \frac{S_T}{2^{10}}\right),$$

which means that (6) holds. $\qquad\square$

## B  Algorithm for Stochastic Bandit Linear Optimization

In this section, we present an algorithm for stochastic bandit linear optimization, where we assume that $\ell_t$ follows a distribution $D$ over $\mathbb{R}^d$, i.i.d. for $t = 1, 2, \ldots, T$. We denote $\ell^* = \mathbf{E}_{\ell \sim D}[\ell] \in \mathbb{R}^d$ and $\xi_t = \ell_t - \ell^*$.

### B.1  Preliminary: Barycentric Spanner

*Definition* 1. Let $S \in \mathbb{R}^d$ be a subset whose linear span is $\mathbb{R}^d$, and let $C > 1$. A set $X = \{x_1, \ldots, x_d\} \subseteq S$ is a *C-barycentric spanner* for $S$ if every $x \in S$ may be expressed as a linear combination of elements of $X$ using coefficients in $[-C, C]$.

**Theorem 6 (Proposition 2.4. in [9]).** *Suppose $\mathcal{P} \subseteq \mathbb{R}^d$ is a compact set not contained in any proper linear subspace. Given an algorithm OPT for LP, for any $C > 1$ we may compute a C-barycentric spanner for $\mathcal{P}$ in polynomial time, using $O(d^2\log_C(d))$ calls to OPT. Its span is equal to $\mathbb{R}^d$.*

### B.2  Algorithm

Our algorithm is summarized in Algorithm 2. In the algorithm, a parameter $\delta > 0$ controls the probability of achieving a small regret. The rounds are divided into $K = O(\log T)$ *phases*, so that the $k$-th phase consists of $\Theta(2^k)$ rounds for each $k \in \{1, \ldots, K\}$.

When the $k$-th phase begins, the algorithm maintains an action set $\mathcal{P}_k$. This action set is initialized by $\mathcal{P}_1 = \mathrm{Conv}(\mathcal{A})$ and is defined recursively by (20) (thus $\mathcal{P}_1 \supseteq \mathcal{P}_2 \supseteq \cdots \supseteq \mathcal{P}_K$). $\mathcal{P}_k$ is designed so that the value of $\ell^{*\top}x$ is smaller for all $x \in \mathcal{P}_k$ as $k$ gets larger (see Lemma 7). At the beginning of the $k$-th phase, the algorithm computes a 2-barycentric spanner $X_k = \{x_{k1}, \ldots, x_{kd}\}$ of $\mathcal{P}_k$. We can construct a good estimate of $\ell^*$ if each element of $X_k$ can be chosen as an action. However, elements in $X_k$ do not always belong to $\mathcal{A}$, especially when $\mathcal{A}$ is discrete. To address this issue, our algorithm decomposes each $x_{ki}$ into the points $x_{ki0}, \ldots, x_{kid} \in \mathcal{A}$ with weight $\lambda_{ki0}, \ldots, \lambda_{kid} \geq 0$ so that $\lambda_{ki0} + \cdots + \lambda_{kid} = 1$ and $\lambda_{ki0}x_{ki0} + \cdots + \lambda_{kid}x_{kid} = x_{ki}$. Then, it plays $x_{kij}, T_{kij} \propto \lambda_{kij}$ times, for each $j = 0, 1, \ldots, d$. We denote the action played at the $t$-th round by $a_t$. The algorithm computes an empirical estimate $\hat{\ell}_k$ of $\ell^*$, based on the feedback obtained in $k$-th phase, as defined in (19).

We note that $\mathrm{Vol}(\mathcal{P}_k) > 0$ holds for all $k$, which implies that we can apply the algorithm in Theorem 6 to $\mathcal{P}_k$ and that $V_k$ is invertible. In fact, we have $\mathrm{Vol}(\mathcal{P}_1) > 0$ from Assumption 1 and we can show

---

**Algorithm 2** An oracle efficient algorithm for stochastic bandit linear optimization

---

**Require:** Action set $\mathcal{A} \subseteq \mathbb{R}^d$, positive real numbers $L$ and $R$ satisfying Assumption 1, $\delta \in (0, 1)$.
1: Set $\mathcal{P}_1 = \text{Conv}(\mathcal{A})$ and $t_{1,1,0} = 0$.
2: **for** $k = 1, 2, \ldots, K$ **do**
3:     Let $X_k = \{x_{k1}, \ldots, x_{kd}\} \subseteq \mathcal{P}_k$ be a 2-barycentric spanner for $\mathcal{P}_k$.
4:     Set $\zeta_k = 2^{2k+9} d^2 \log\left(\frac{2dk(k+1)}{\delta}\right)$.
5:     **for** $i = 1, \ldots, d$ **do**
6:         Solve DP for $\mathcal{P} = \text{Conv}(\mathcal{A})$ and $x = x_{ki}$ to get a decomposition $x_{ki0}, \ldots, x_{kid} \in \mathcal{A}$ and $\lambda_{ki0}, \ldots, \lambda_{kid}$ such that $x_{ki} = \lambda_{ki0} x_{ki0} + \cdots + \lambda_{kid} x_{kid}$.
7:         **for** $j = 0, \ldots, d$ **do**
8:             Set $T_{kij} = \lceil \zeta_k \lambda_{kij} \rceil, t_{ki,j+1} = t_{kij} + T_{kij}$.
9:             Choose action $a_t = x_{kij}$ exactly $T_{kij}$ times, from the $(t_{kij} + 1)$-th round to the $(t_{ki,j+1})$-th round.
10:        **end for**
11:        Set $t_{k,i+1,0} = t_{ki,d+1}$.
12:    **end for**
13:    Calculate empirical estimate $\hat{\ell}_k$ of $\ell^*$ by

$$V_k = \sum_{i=1}^{d} \sum_{j=0}^{d} T_{kij} x_{kij} x_{kij}^\top, \quad \hat{\ell}_k = V_k^{-1} \sum_{i=1}^{d} \sum_{j=0}^{d} \sum_{t=t_{kij}+1}^{t_{ki,j+1}} (\ell_t^\top x_{kij}) x_{kij}. \tag{19}$$

14:    Solve LP for $\mathcal{P} = \mathcal{P}_k$ and $w = \hat{\ell}_k$ to find a vector $x_k^* \in \underset{x \in \mathcal{P}_k}{\arg\min} \, \hat{\ell}_k^\top x$.
15:    Update $\mathcal{P}_k$ by

$$\mathcal{P}_{k+1} = \{x \in \mathcal{P}_k \mid \hat{\ell}_k^\top (x - x_k^*) \leq LR2^{-k}\}. \tag{20}$$

16: **end for**

---

$\text{Vol}(\mathcal{P}_k) > 0$ by induction in $k$, from the definition (20). The linear span of a barycentric spanner $X_k = \{x_{k1}, \ldots, x_{kd}\}$ coincides with that of $\mathcal{P}_k$ (see, e.g., [9]), which is equal to $\mathbb{R}^d$, because $\text{Vol}(\mathcal{P}_k) > 0$. Hence, we have $V_k \succeq \zeta_k \sum_{i=1}^{d} \sum_{j=0}^{d} \lambda_{kij} x_{kij} x_{kij}^\top \succeq \zeta_k \sum_{i=1}^{d} x_{ki} x_{ki}^\top \succ O$, which means that $V_k$ is nonsingular.

Algorithm 2 satisfies the following regret bound:

**Theorem 7.** *Suppose that $\ell_t$ follows an i.i.d. distribution for $t = 1, \ldots, T$ with $T \geq 2$, and that $\{a_t\}_{t=1}^{T}$ is given by Algorithm 2. With probability at least $1 - \delta$, the regret is bounded as follows:*

$$\max_{a \in \mathcal{A}} R_T(a) \leq 2^{12} LR \sqrt{d^3 T \log\left(\frac{d \log T}{\delta}\right)}. \tag{21}$$

The proof of this theorem is given in Appendix C.

### B.3 Oracle Complexity Analysis

Step 3 in Algorithm 2 requires constructing a 2-barycentric spanner. From Theorem 6, we can construct it by calling an algorithm that solves LP for $\mathcal{P} = \mathcal{P}_k$, $O(\text{poly}(d))$ times. Theorem 3 (a) and Remark 1 imply that we can solve LP by solving SP $O(\text{poly}(d, \log T))$ times. SP for $\mathcal{P} = \mathcal{P}_k$ can be solved by the following procedure:

1. Decide if $y \in \mathcal{P}_1$ or not, and, in the latter case, output a vector $w \in \mathbb{R}^d$ such that $w^\top y < \min_{x \in \mathcal{P}_1} w^\top x$. From Theorem 3 (b), this can be done by calling the LP oracle for $\mathcal{A}$ $O(\text{poly}(d))$ times. In the former case, i.e., if $y \in \mathcal{P}_1$, go to the next step.

2. For $j = 1, \ldots, k - 1$, if $\hat{\ell}_j^\top (y - x_k^*) > LR2^{-j}$, output $\hat{\ell}_j$. If $\hat{\ell}_j^\top (y - x_k^*) \leq LR2^{-j}$ for all $j = 1, \ldots, k - 1$, it means that $y \in \mathcal{P}_k$.

This procedure calls the LP oracle for $O(\mathrm{poly}(d, \log T))$ times and runs in $O(\mathrm{poly}(d, K)) = O(\mathrm{poly}(d, \log T))$ times. Hence, we have an efficient implementation of Step 3 that calls the LP oracle for $\mathcal{A}$ $O(\mathrm{poly}(d, \log T))$ times. Similarly, Steps 6 and 14 in Algorithm 2 can be executed by calling the LP oracle for $\mathcal{A}$ $O(\mathrm{poly}(d, \log T))$ times. The other steps are free from access to the oracle and can be efficiently implemented. Because the number $K$ of iterations w.r.t. $k$ is bounded as in (32), the number of oracle calls for solving LP over $\mathcal{A}$ is of $O(\mathrm{poly}(d, \log T)K) = O(\mathrm{poly}(d, \log T))$.

## C    Regret Analysis of Algorithm 2

In the proof of Theorem 7, we may assume that

$$T > 2d\zeta_1 = 2^{12}d^3 \log\left(\frac{4d}{\delta}\right). \tag{22}$$

Indeed, if $T \le 2d\zeta_1$, then we see $R_T(a) \le 2LRT \le 2LR\sqrt{2d\zeta_1 T}$, which means that (21) holds.

To prove the above theorem, we start with analyzing the error of the estimators $\hat{\ell}_k$ defined by (19):

**Lemma 6.** *With probability at least $1 - \delta$, for all $k \in \{1, 2, \ldots\}$ and $x \in \mathcal{P}_k$, we have*

$$|(\hat{\ell}_k - \ell^*)^\top x| \le 2^{-1-k}LR. \tag{23}$$

*Proof.* Because $X_k = \{x_{k1}, \ldots, x_{kd}\}$ is a 2-barycentric spanner for $\mathcal{P}_k$, for all $x \in \mathcal{P}_k$, there exists a vector $w = (w_1, \ldots, w_d)^\top \in [-2, 2]^d$ such that $x = w_1 x_{k1} + \cdots + w_d x_{kd}$. By means of this $w$, $(\hat{\ell}_k - \ell^*)^\top x$ can be expressed as

$$
\begin{aligned}
(\hat{\ell}_k - \ell^*)^\top x &= \left( \sum_{i=1}^{d}\sum_{j=0}^{d}\sum_{t=t_{kij}+1}^{t_{ki,j+1}} \ell_t^\top x_{kij} x_{kij}^\top V_k^{-1} - \ell^{*\top} \right) \sum_{s=1}^{d} w_s x_{ks} \\
&= \left( \sum_{i=1}^{d}\sum_{j=0}^{d}\sum_{t=t_{kij}+1}^{t_{ki,j+1}} (\ell_t - \ell^*)^\top x_{kij} x_{kij}^\top V_k^{-1} \right) \sum_{s=1}^{d} w_s x_{ks} \\
&= \sum_{s=1}^{d} w_s \sum_{i=1}^{d}\sum_{j=0}^{d}\sum_{t=t_{kij}+1}^{t_{ki,j+1}} (\xi_t^\top x_{kij}) x_{kij}^\top V_k^{-1} x_{ks},
\end{aligned} \tag{24}
$$

where the first equality comes from the definition (19) of $\hat{\ell}_k$, the second equality comes from $\sum_{i=1}^{d}\sum_{j=0}^{d}\sum_{t=t_{kij}+1}^{t_{ki,j+1}} x_{kij} x_{kij}^\top V_k^{-1} = \sum_{i=1}^{d}\sum_{j=0}^{d} T_{kij} x_{kij} x_{kij}^\top V_k^{-1} = I$, and the last equality comes from $\xi_t = \ell_t - \ell^*$. We give a uniform bound for this value by the following claim: with probability at least $1 - \delta$, it holds for all $k = 1, 2, \ldots$ and $s = 1, \ldots, d$ that

$$\left| \sum_{i=1}^{d}\sum_{j=0}^{d}\sum_{t=t_{kij}+1}^{t_{ki,j+1}} (\xi_t^\top x_{kij}) x_{kij}^\top V_k^{-1} x_{ks} \right| \le \frac{LR}{d2^{k+2}}. \tag{25}$$

Because the expectation of $\xi_t = \ell_t - \ell^*$ is equal to 0, and because $\|\xi_t\|_2 \le \|\ell_t\|_2 + \|\ell^*\|_2 \le 2L$ and $\|x_{kij}\|_2 \le R$ hold from Assumption 1, $\{\xi_t^\top x_{kij}\}$ are independent random variables with mean 0 and absolute value at most $2LR$. Hence, from Hoeffding's inequality, it holds with probability at least

$1 - \frac{\delta}{dk(k+1)}$ that

$$\left| \sum_{i=1}^{d} \sum_{j=0}^{d} \sum_{t=t_{kij}+1}^{t_{ki,j+1}} (\xi_t^\top x_{kij}) x_{kij}^\top V_k^{-1} x_{ks} \right|$$

$$\leq 2LR \sqrt{8 \log\left( \frac{2dk(k+1)}{\delta} \right) \sum_{i=1}^{d} \sum_{j=0}^{d} \sum_{t=t_{kij}+1}^{t_{ki,j+1}} (x_{kij}^\top V_k^{-1} x_{ks})^2}$$

$$= 2LR \sqrt{8 \log\left( \frac{2dk(k+1)}{\delta} \right) x_{ks}^\top V_k^{-1} \left( \sum_{i=1}^{d} \sum_{j=0}^{d} \sum_{t=t_{kij}+1}^{t_{ki,j+1}} x_{kij} x_{kij}^\top \right) V_k^{-1} x_{ks}}$$

$$= 2LR \sqrt{8 \log\left( \frac{2dk(k+1)}{\delta} \right) x_{ks}^\top V_k^{-1} x_{ks}}. \tag{26}$$

The value $x_{ks}^\top V_k^{-1} x_{ks}$ is bounded as $x_{ks}^\top V_k^{-1} x_{ks} \leq \zeta_k^{-1}$. Indeed, we have

$$V_k \succeq \sum_{j=0}^{d} T_{ksj} x_{ksj} x_{ksj}^\top \succeq \zeta_k \sum_{j=0}^{d} \lambda_{ksj} x_{ksj} x_{ksj}^\top$$

$$= \zeta_k \left( x_{ks} x_{ks}^\top + \sum_{j=0}^{d} \lambda_{ksj} (x_{ksj} - x_{ks})(x_{ksj} - x_{ks})^\top \right) \succeq \zeta_k x_{ks} x_{ks}^\top,$$

where the first inequality comes from the definition (19) of $V_k$, the second inequality comes from $T_{ksj} = \lceil \zeta_k \lambda_{ksj} \rceil \geq \zeta_k \lambda_{ksj}$ (Step 8 of Algorithm 2), and the equality comes from that $\lambda_{ks0} + \cdots + \lambda_{ksd} = 1$ and that $\lambda_{ks0} x_{ks0} + \cdots + \lambda_{ksd} x_{ksd} = x_{ks}$. This inequality indicates

$$0 \leq (V_k^{-1} x_{ks})^\top (V_k - \zeta_k x_{ks} x_{ks}^\top)(V_k^{-1} x_{ks}) = x_{ks}^\top V_k^{-1} x_{ks} (1 - \zeta_k x_{ks}^\top V_k^{-1} x_{ks}),$$

from which we have $x_{ks}^\top V_k^{-1} x_{ks} \leq \zeta_k^{-1}$.

Plugging this bound on $x_{ks}^\top V_k^{-1} x_{ks}$ into (26) and the definition of $\zeta_k$ (Step 4 of Algorithm 2) show that the inequality (25) holds with probability at least $1 - \frac{\delta}{dk(k+1)}$ for each $k$ and $s$. Hence, we have

$$\text{Prob}[\,(25) \text{ does not hold for some } k \text{ and } s\,] \leq \sum_{k=1}^{K} \sum_{s=1}^{d} \text{Prob}[\,(25) \text{ does not hold for } k \text{ and } s\,]$$

$$\leq \sum_{k=1}^{K} \sum_{s=1}^{d} \frac{\delta}{dk(k+1)} = \delta,$$

which means that, with probability at least $1 - \delta$, (25) holds for all $k$ and $s$. Combining this, (24), and $|w_s| \leq 2$ for all $s \in [d]$, we obtain (23). □

**Lemma 7.** *Fix $a^* \in \arg\min\limits_{a \in \mathcal{A}} \ell^{*\top} a$. With probability at least $1 - \delta$, for all $k$, we have*

$$a^* \in \mathcal{P}_k, \text{ and } \ell^{*\top} x - \ell^{*\top} a^* \leq 2^{2-k} LR \text{ for all } x \in \mathcal{P}_k. \tag{27}$$

*Proof.* From Lemma 6, we can assume that (23) holds for all $k$ and $x \in \mathcal{P}_k$. Under this assumption, we show (27) by induction in $k$. We can confirm that (27) holds for $k = 1$. Indeed, $a^* \in \mathcal{P}_1$ follows from $\mathcal{A} \subseteq \mathcal{P}_1$, and $\ell^{*\top} x - \ell^{*\top} a^* \leq \|\ell^*\|_2 \|x - a^*\|_2 \leq 2LR$ follows from $\|\ell^*\|_2 \leq L$ and $\|x\|_2 \leq R$ for $x \in \mathcal{P}_1$. Suppose that (27) holds for $k = s$. Then, $\mathcal{P}_{s+1}$, defined by (20), contains $a^*$ because

$$\hat{\ell}_s^\top a^* \leq \ell^{*\top} a^* + 2^{-1-s} LR \leq \ell^{*\top} x_s^* + 2^{-1-s} LR \leq \hat{\ell}_s^\top x_s^* + 2^{-s} LR,$$

where the first and the third inequalities come from (23) and the second inequality comes from $\ell^{*\top} a^* = \min_{a \in \mathcal{A}}\{\ell^{*\top} a\} = \min_{x \in \mathcal{P}_1}\{\ell^{*\top} x\} \leq \ell^{*\top} x_s^*$. Furthermore, for all $x \in \mathcal{P}_{s+1}$, we have

$$\ell^{*\top} x \leq \hat{\ell}_s^\top x + 2^{-s-1} LR \leq \hat{\ell}_s^\top x_s^* + 3 \cdot 2^{-s-1} LR \leq \hat{\ell}_s^\top a^* + 3 \cdot 2^{-s-1} LR \leq \ell^{*\top} a^* + 2^{-s+1} LR,$$

where the first and the fourth inequalities come from (23), the second inequality comes from the definition (20) of $\mathcal{P}_{k+1}$, and the third inequality comes from $a^* \in \mathcal{P}_s$ and $x_s^* \in \arg\min_{x \in \mathcal{P}_s} \hat{\ell}_s^\top x$. Hence, (27) holds for $k = s + 1$. By induction in $k$, (27) is proven to hold for all positive integers $k$. $\quad\square$

From this lemma and the definition of $a_t$ in Algorithm 2, we can bound $\sum_{t=t_{k,1,0}+1}^{t_{k+1,1,0}} \ell^{*\top}(a_t - a^*)$ as follows:

$$\sum_{t=t_{k,1,0}+1}^{t_{k+1,1,0}} \ell_t^{*\top}(a_t - a^*) = \sum_{i=1}^{d}\sum_{j=0}^{d} T_{kij}\ell^{*\top}(x_{kij} - a^*) \leq \sum_{i=1}^{d}\sum_{j=0}^{d}(\zeta_k\lambda_{kij}\ell^{*\top}(x_{kij} - a^*) + 2LR)$$

$$= \sum_{i=1}^{d}(\zeta_k\ell^{*\top}(x_{ki} - a^*) + 2LR(d+1)) \leq dLR(2^{2-k}\zeta_k + 2(d+1)),$$

$$(28)$$

where the first inequality follows from $T_{kij} = \lceil \zeta_k\lambda_{kij} \rceil$ (Step 8 of Algorithm 2) and $|\ell^{*\top}(x_{kij} - a^*)| \leq 2LR$, the second inequality follows from $\sum_{j=0}^{d} \lambda_{kij}x_{kij} = x_{ki}$ and $\sum_{j=0}^{d} \lambda_{kij} = 1$, and the last inequality follows from Lemma 7.

Let $T_k$ denote the number of rounds in the $k$-th phase, i.e.,

$$T_k = \sum_{i=1}^{d}\sum_{j=0}^{d} T_{kij} = t_{k+1,1,0} - t_{k,1,0}. \qquad (29)$$

From the definition of $T_{kij} = \lceil \zeta_k\lambda_{kij} \rceil$ (Step 8 of Algorithm 2), we have $\zeta_k\lambda_{kij} \leq T_{kij} < \zeta_k\lambda_{kij} + 1$. Combining this and the condition that $\sum_{j=0}^{d} \lambda_{kij} = 1$, we obtain

$$d\zeta_k \leq T_k \leq d\zeta_k + d(d+1). \qquad (30)$$

Let $K$ be the index of phases such that the $K$-th phase includes $T$-th round, i.e., $K$ is the number such that $\sum_{k=1}^{K-1} T_k < T \leq \sum_{k=1}^{K} T_k$. Note that $K \geq 2$ follows from the assumption 22. From (30) and the definition of $\zeta_k$ (Step 4 in Algorithm 2), we have

$$T > T_{K-1} \geq d\zeta_{K-1} = 2^{2K+7}d^3\log\left(\frac{2dK(K-1)}{\delta}\right) \geq (2^{K+3})^2d^3\log\left(\frac{2dK(K+1)}{\delta}\right),$$

where the last inequality follows from $2\log\left(\frac{2dK(K-1)}{\delta}\right) = \log\left(\frac{2dK(K-1)}{\delta}\right)^2 \geq \log\left(\frac{2dK(K+1)}{\delta}\right)$. This inequality implies the bound on $2^K$ and $K$, as follows:

$$2^K \leq \frac{1}{2^3}\left(\frac{T}{d^3}\right)^{\frac{1}{2}}\left(\log\left(\frac{2dK(K+1)}{\delta}\right)\right)^{-\frac{1}{2}}, \qquad (31)$$

$$K \leq \frac{1}{2}\log_2 T - 3. \qquad (32)$$

By means of these inequalities, we can bound the value $\sum_{t=1}^{T} \ell^{*\top}(a_t - a^*)$, for the output $a_t$ of Algorithm 2, and $a^* \in \arg\min_{a \in \mathcal{A}} \ell^{*\top} a$:

$$
\begin{aligned}
\sum_{t=1}^{T} \ell^{*\top}(a_t - a^*) &\leq \sum_{k=1}^{K} \sum_{t=t_{k,1,0}+1}^{t_{k+1,1,0}} \ell^{*\top}(a_t - a^*) \\
&\leq dLR \sum_{k=1}^{K} (2^{2-k}\zeta_k + 2(d+1)) && \text{(from (28))} \\
&= dLR \sum_{k=1}^{K} \left( 2^{k+11} d^2 \log\left(\frac{2dk(k+1)}{\delta}\right) + 2K(d+1) \right) && \text{(Step 4 in Algo. 2)} \\
&\leq dLR \left( 2^{K+12} d^2 \log\left(\frac{2dK(K+1)}{\delta}\right) + 2K(d+1) \right) \\
&\leq dLR \left( 2^9 d^2 \left( \frac{T}{d^3} \log\left(\frac{2dK(K+1)}{\delta}\right) \right)^{\frac{1}{2}} + 2K(d+1) \right) && \text{(from (31))} \\
&\leq dLR \left( 2^9 \sqrt{dT \log\left(\frac{d(\log_2 T)^2}{\delta}\right)} + (d+1)\log_2 T \right) && \text{(from (32))} \\
&\leq dLR \left( 2^{10} \sqrt{dT \log\left(\frac{d \log T}{\delta}\right)} + (d+1)\log_2 T \right). && (33)
\end{aligned}
$$

By combining this and the following lemma, we obtain an upper bound on the regret $R_T(a) = \sum_{t=1}^{T} \ell_t^{\top}(a_t - a)$.

**Lemma 8.** *Let $a^* \in \arg\min_{a \in \mathcal{A}} \ell^{*\top} a$. With probability at least $1 - \delta$, it holds for all $a \in \mathcal{A}$ that*

$$
R_T(a) \leq \sum_{t=1}^{T} \ell^{*\top}(a_t - a^*) + 8LR\sqrt{dT \log\left(\frac{2d}{\delta}\right)}. \tag{34}
$$

*Proof.* We show (34) by proving the following two inequalities:

$$
\sum_{t=1}^{T} \ell_t^{\top} a_t - \sum_{t=1}^{T} \ell^{*\top} a_t \leq LR\sqrt{8T \log\left(\frac{2}{\delta}\right)}, \tag{35}
$$

$$
\sum_{t=1}^{T} \ell^{*\top} a^* - \sum_{t=1}^{T} \ell_t^{\top} a \leq LR\sqrt{8dT \log\left(\frac{2d}{\delta}\right)}. \tag{36}
$$

Denote $X_\tau := \sum_{t=1}^{\tau} (\ell_t - \ell^*)^{\top} a_t$. Because $\{X_\tau\}_{\tau=0}^{T}$ is a martingale such that $|X_{\tau+1} - X_\tau| \leq 2LR$, from Azuma's inequality, with probability $1 - \frac{\delta}{2}$, we have $X_T \leq LR\sqrt{8T \log\left(\frac{2}{\delta}\right)}$, which means that (35) holds. Similarly, from Hoeffding's inequality, we have

$$
\left\| \sum_{t=1}^{T} (\ell_t - \ell^*) \right\|_2 \leq L\sqrt{dT \log\left(\frac{2d}{\delta}\right)} \tag{37}
$$

with probability at least $1 - \frac{\delta}{2}$. Under this condition, we have

$$
-\sum_{t=1}^{T} \ell_t^{\top} a \leq -\sum_{t=1}^{T} \ell^{*\top} a + LR\sqrt{dT \log\left(\frac{2d}{\delta}\right)} \leq -\sum_{t=1}^{T} \ell^{*\top} a^* + LR\sqrt{8dT \log\left(\frac{2d}{\delta}\right)},
$$

where the first inequality follows from (37) and $\|a\|_2 \leq R$, and the second inequality follows from $a^* \in \arg\min_{a \in \mathcal{A}} \ell^{*\top} a$. Hence, we have (36) for all $a \in \mathcal{A}$ with probability at least $1 - \frac{\delta}{2}$. Because each of (35) and (36) holds with probability $1 - \frac{\delta}{2}$, both (35) and (36) hold with probability $1 - \delta$. Then, by taking the sum of each side of (35) and (36), we obtain (34). □

By combining (33), Lemma 8 and (22), we obtain

$$R_T(a) \leq 2^{11} LR \sqrt{d^3 T \log\left(\frac{d \log T}{\delta}\right)}$$

with probability at least $1 - 2\delta$. Replacing $\delta$ with $\delta/2$, we obtain (21).