[Reviews · NeurIPS 2019]

Reviewer 1



The paper considers the online linear optimization problem with bandit feedback in both adversarial and stochastic settings. The paper aims at constructing algorithms with $\tilde{O}(\sqrt{T})$ regret guarantee and low oracle-complexity, where the considered oracle solves linear optimization problems on the set of actions. The proposed algorithms achieve $\tilde{O}(T)$ oracle-complexity in the non-stochastic case and $\tilde{O}(\operatorname{poly}(d,\log T))$ in the stochastic case. These results are a significant progress with respect to existing work. Besides, the construction and analysis of the algorithms are highly non-trivial. I was not able to check all the details of the proofs, but I have no reason to doubt their correctness. Such a strong focus on the complexity of the algorithms in this problem is novel (as far as I know) and very interesting. Overall, I feel that this paper is high quality work.

Reviewer 2



This paper propose a new method for sampling in the procedure of choosing an arm to pull in both stochastic setting and non-stochastic setting. Their algorithms reduce the oracle complexity a lot, while maintaining the regret upper bound the same. The method is a good approach, and as the authers mentioned, it can reduce the complexity a lot. However, I am not sure whether it is an important problem. In this paper, the authors does not list lots of references about this. I think there need to be more explainations. I do not check the proofs in detail. I can understand the analysis from the intuition idea they provided, and they seems to be correct. In the rebuttal, the authors show more related works about this problem, and lists why their work is important. I think such explanations are convincing, and I suggest the authors to add these explanations into their paper, so that their contributions are clearer to the readers who are not so familiar with this problem.

Reviewer 3



This paper gives the first algorithm that has O(\sqrt{T}) regret and linear oracle complexity for non-stochastic setting and sublinear oracle complexity for stochastic setting. Originality: It extends the cutting-plane approach from full-information setting to bandit setting using the well-known relations between linear optimization, seperation and decomposition on convex body. Quality: The proofs are reliable. Clarity: This paper gives a thorough introduction about the known method on this area and gives a clear comparison between the known ones and the one they propose. Clearly written. Significance: The algorithm in this paper is important because it maintains a low regret and it is efficient, unlike other known low regret algorithms which are inefficient to apply.

[Author Response · NeurIPS 2019]

Dear Reviewer #1:

> numerical experiments benchmarking both regret minimization and computing time

Thanks for your comments. We agree with the idea that numerical experiments would be beneficial. Empirical
comparison with previous algorithms is an important future task.

5
6

Dear Reviewer #2:

> the authors does not list lots of references about this. I think there need to be more explainations.

> Some explanation about existing works about reducing the oracle complexity, or show some evidences
about the importance of this problem.

Reducing oracle complexity is practically important as the computational efficiency of algorithms heavily depends on
the number of oracle calls, especially when each oracle call for linear optimization takes much time. As mentioned in
[20], small oracle complexities imply that an offline algorithm is sufficient to obtain small regrets in online decision
problems with bandit feedback, i.e., there is an efficient algorithm for offline-to-online conversion. Furthermore,
analyzing oracle complexities has theoretical significance because it gives insights on computational complexity theory.
For example, polynomial oracle complexities may imply that each bandit optimization problem belongs to the same
complexity class as the corresponding offline optimization problem, under polynomial-time reductions. Existing works
about reducing the oracle complexity is reviewed in the following:

There are many works focusing on the oracle complexity of online linear optimization problems with *full information*,
in which a player can observe all entries of the loss vector $\ell_t$. Kalai and Vempara [22] reduced the oracle complexity
into $O(T)$ while keeping $O(\sqrt{T})$ regret bound. As an extension of this result, the online improper learning setting
has been considered. Online improper learning is a generalized framework of online optimization, in which only an
approximate offline optimization oracle is given, and the performance of online algorithms is evaluated by means of the
approximate regret. For this problem, Kakade et al. [21] proposed an algorithm that achieves an approximate regret
bound of $O(\sqrt{T})$ with an oracle complexity of $\tilde{O}(T^2)$. Later, Garber [18] and Hazan et al. [20] reduced the oracle
complexity into $\tilde{O}(T^{3/2})$ and $\tilde{O}(T)$, respectively.

For the problems with *bandit feedback*, most existing works about reducing the oracle complexity sacrifice regret
bounds and suffer regrets of $\tilde{O}(T^{2/3})$, a suboptimal rate. Such results are given by converting the algorithms for
full-information settings to those for bandit-feedback settings. For example, the result of Kalai and Vempara [22] for
the full-information setting is extended to the bandit-feedback setting by [Dani and Hayes, [15]] and [McMahan and
Blum, [25]] to give an algorithm that achieves a regret bound of $\tilde{O}(T^{2/3})$ and an oracle complexity of $O(T^{2/3})$. For
online improper learning with bandit feedback, there have been similar results achieving approximate regret bounds
of $\tilde{O}(T^{2/3})$ with oracle complexity of $\tilde{O}(\text{poly}(T))$ (Kakade et al. [21]), $\tilde{O}(T)$ (Garbar [18]) and $\tilde{O}(T^{2/3})$ (Hazan et
al. [20]). Achieving $\tilde{O}(T^{1/2})$ regret for bandit improper learning with small oracle complexity has been mentioned as
an open question in literature such as [20] and [21].

In the revised version, we will modify the section of related works to highlight existing results on oracle complexities
and to clarify the importance of oracle complexities.

38
39

Dear Reviewer #3:

> It would be nice if they could also mention the general bandit problem and give some comments on whether
they can extend their method there.

Thanks for your comments. We may consider two generalizations: nonlinear convex bandits [10] and bandit online
improper learning [18, 20, 21]. However, there seem to be difficulties in extending our methods to these problems.
To extend our results to nonlinear convex bandits, we need to construct estimates of gradients. In the nonlinear
case, estimated gradients include biases, which makes the problem and the analysis complicated. To the improper
learning setting, our approach cannot be directly applied because solving the separation problem is hard when only an
approximate oracle is given.

> The paper would be even more persuasive if they can provide some numerical results.

Please take a look at the response to reviewer #1.

[Meta-Review · NeurIPS 2019]

Nice paper! Please take the reviewers suggestions into consideration when finalising your paper.